

**Technical Note: Nighttime OH and HO$_2$ chemical equilibria in the mesosphere – lower**
**thermosphere**
Mikhail Yu. Kulikov[1], Mikhail V. Belikovich[1], Aleksey G. Chubarov[1], Svetlana O. Dementyeva[1], and
Alexander M. Feigin[1]
[1]A. V. Gaponov-Grekhov Institute of Applied Physics of the Russian Academy of Sciences, 46 Ulyanov
Str., 603950 Nizhny Novgorod, Russia
Correspondence to: Mikhail Yu. Kulikov (mikhail_kulikov@mail.ru)
**Abstract.** At the altitudes of the mesosphere – lower thermosphere, OH and HO$_2$ play a significant
role in many physicochemical processes. Thus, monitoring of their spatiotemporal evolution together with
other chemically active trace gases is one of the most important problems for this atmosphere region, in
which direct measurements are difficult. The paper studies the nighttime OH and HO$_2$ chemical equilibria
using the 3D chemical transport modeling within the general approach including the extraction of the
main sources and sinks in the equilibrium space-time areas and derivation of analytical criteria for
equilibrium validity. The presented analysis shows that there are extended areas, where nighttime HO$_2$
and OH are close to their local equilibrium concentrations determined mainly by the reaction between
HO$_x$ – O$_x$ components themselves and with H$_2$O$_2$, N, NO, NO$_2$, and CO. In the upper mesosphere – lower
thermosphere, the equilibrium expressions can be shortened, including the HO$_x$ – O$_x$ chemistry only.
These conditions describes the HO$_2$ and OH equilibrium from the top to the some lower borders, the
altitude position of which vary in the interval between 72-73 and 85 km and depends essentially on
season and latitude. The developed analytical criteria almost everywhere well reproduce the main features
of these borders. The obtained results allow to extend the abilities of previously proposed methods for the
retrieval of poorly measured components from measurement data and to develop new approaches.



## 1 Introduction

Monitoring the spatiotemporal evolution of chemically active trace gases is one of the most important problems in atmospheric research. Despite the increase of the experimental data volume nowadays, primarily due to the development of remote sensing methods, many important trace gases continue to be unavailable for direct and regular measurements. A well-known way to increase the information content of experimental campaigns is to use chemical transport models and available experimental data for deriving unmeasured characteristics indirectly. Within the framework of this approach, the model acts as *a priori* relationship between directly measured and retrieved characteristics. The simplest model, that makes it possible to implement this approach, is based on the condition of local (in both time and space) photochemical/chemical balance (local equilibrium) between sources and sinks of the so-called "fast" components: trace gases with short lifetimes relatively, in particular, to the characteristic transport times. Mathematically, this condition does not mean that the fast variables are at equilibrium, but when it is fulfilled, the corresponding concentrations are close to their instantaneous equilibrium values. At the same time, due to the strong dissipation, in most cases (except the special cases where the ensemble of fast components includes the slow family of these components), there is no need to follow the law of matter conservation. It is possible to discard insignificant sinks and sources in the corresponding balance equations without loss of accuracy, including those caused by transport. The resulting algebraic equations are the simplest a priori local relations between measurable and retrieved trace gases. These relationships can be used to derive information about hard-to-measure atmospheric species, determine key atmospheric characteristics (for example, temperature), validate the data quality of simultaneous measurements of several atmospheric components, estimate reaction rate constants known with high uncertainty, evaluate sources (emissions), etc.

For several decades, the photochemical/chemical equilibrium approximation have been used to solve many atmospheric tasks. It is applied (see, e.g., the short review in Kulikov et al. (2018a) and references therein) in investigations of the surface layer and free troposphere chemistry in different regions (over megalopolises, in rural areas, in the mountains, over the seas), in stratospheric chemistry studies, including derivation of a critical parameters of the ozone destruction catalytic cycles, and to study the $HO_x - O_x$ chemistry and airglows ($O(^1S)$ green-line, $O_2$ A-band, OH Meinel band emissions) at the heights of the mesosphere – lower thermosphere. In the latter case, the distributions of unmeasured characteristics are determined from the data of daytime and nighttime rocket and satellite measurements (e.g., Evans and Llewellyn, 1973; Good, 1976; Pendleton et al., 1983; McDade et al., 1985; McDade and Llewellyn, 1988; Evans et al., 1988; Thomas, 1990; Llewellyn et al., 1993; Llewellyn and McDade, 1996; Russell and Lowe, 2003; Russell et al., 2005; Kulikov et al., 2006, 2009, 2017, 2022a, 2022b; Mlynczak et al., 2007, 2013a, 2013b, 2014, 2018; Smith et al., 2010; Xu et al., 2012; Siskind et al., 2008, 2015;





Fytterer et al., 2019) with the use of equilibrium assumptions for ozone and excited states of OH, O, and
$O_2$. For example, such an approach is applied to the data of the SABER (Sounding of the Atmosphere
using Broadband Emission Radiometry) instrument onboard the TIMED (Thermosphere Ionosphere
Mesosphere Energetics and Dynamics) satellite, which since 2002 continues to measure simultaneous
profiles of temperature, ozone, and volume emission rates of OH* transitions in wide ranges of altitude,
local time, and latitude with a rather high space-time resolution.
Note a number of general aspects of the application of equilibrium conditions in the above
examples. First, there are no clear criteria why the equilibrium condition should be satisfied. Usually, a
certain component is taken to be a fast variable if its lifetime is much shorter than the lifetimes of other
components of studied photochemical/chemical system or the duration of a day, daytime, nighttime, etc.
For example, in the papers on SABER data processing (Mlynczak et al., 2013a, 2013b, 2014, 2018), it is
assumed that the nighttime ozone chemical equilibrium in the mesopause is well fulfilled at altitudes of
80–100 km, since the nighttime ozone lifetime at these altitudes is too short varying in the range from
several minutes to several tens of minutes. Note this assumption is quite popular and used in different
tasks (e.g., Swenson and Gardner, 1998; Marsh et al., 2006; Smith et al., 2009; Nikoukar et al., 2007; Xu
et al., 2010, 2012; Kowalewski et al., 2014; Grygalashvyly et al., 2014; Grygalashvyly, 2015; Sonnemann
et al., 2015; Kulikov et al., 2021). Belikovich et al. (2018) and Kulikov et al. (2018b, 2019, 2023a)
analyzed the nighttime ozone chemical equilibrium numerically, analytically, and with the use of
SABER/TIMED data. It was revealed that the short lifetime is not a sufficient condition, so this
equilibrium may be significantly disrupted above 80 km. Secondly, there is no detailed numerical
examination of this approximation validity, depending on altitude, latitude, local time, and season.
Correspondingly, there is no the assessment of possible errors in retrieved characteristics due to
disturbance of the used equilibrium condition.
Starting since our papers by Belikovich et al. (2018) and Kulikov et al. (2018b, 2019, 2023a), we
develop the general approach to correct search of fast components using the data from a global 3D
chemical transport model. It includes:
1. Plotting of the equilibrium space-time maps of interested component.
2. Identification of the main sources and sinks in the found equilibrium areas.
3. Derivation and subsequent use of analytical criteria that make it possible to determine the fulfillment of
the equilibrium condition locally (in time and space) with the use of the measurement data only.
The last point is based on the theory of chemical equilibrium of a certain trace gas based on estimations of
its lifetime and equilibrium concentration and time dependences of these characteristics (Kulikov et al.,
2023a).





The main goals of this paper is to apply mentioned approach for analysis of nighttime OH and $HO_2$
chemical equilibriums in the mesosphere – lower thermosphere. Along with O and H, OH and $HO_2$ are
important components of $HO_x$ – $O_x$ chemistry participating (a) in chemical heating through, in particular,
O+OH $\rightarrow$ $O_2$+H and O+$HO_2$ $\rightarrow$ $O_2$+OH exothermic reactions, (b) in formation of airglows, (c) in
catalytic cycles of the ozone destruction. Moreover, the equilibrium conditions of OH and $HO_2$ are
additional *a priori* relationships that can be used to retrieve these components or other characteristics
from measured data. In particular, the method proposed by Panka et al. (2021) for total OH retrieval from
SABER/TIMED data at 80-100 km does not use the nighttime ozone chemical equilibrium, but
nevertheless, applies the equilibrium between sources and sinks for all exited and ground states of OH
(v=0-9). Therefore, this approach is tested in our paper.
In the next section, we present the used model and methods. In Section 3, the model data are used to
plot $HO_2$ and OH equilibrium maps. In Sections 4-5, we extract the main reactions determining equilibria
of these gases and present their shortened equilibria conditions at the upper mesosphere and lower
thermosphere altitudes. In Section 6, the criteria for $HO_2$ and OH equilibria validity are developed. In
Section 7, we discuss the obtained results and their possible applications.

## 2 Used 3D model and Approaches


The analysis of OH and $HO_2$ nighttime chemical equilibria was carried out using the data obtained
with calculation of 3D chemical transport model of the middle atmosphere developed at the Leibniz
Institute of Atmospheric Physics (e.g., Sonnemann et al., 1998; Körner & Sonnemann, 2001;
Grygalashvyly et al., 2009; Hartogh et al., 2004, 2011) to investigate the mesosphere – lower
thermosphere chemistry, in particular, in the extended mesopause region. A number of papers (e.g.,
Hartogh et al., 2004, 2011; Sonnemann, et al., 2006, 2008) validated the model with measurements, in
particular, for ozone and water vapour.
The space-time distribution of temperature and winds were taken from the model of the dynamics
of the middle atmosphere COMMA-IAP (e.g., Kremp et al., 1999; Berger and von Zahn, 1999) with an
updated frequency of 1 day and linear smoothing between subsequent updates to avoid unrealistic jumps
in the calculated concentrations of trace gases. 3D advective transport is taken into account with the use
of the Walcek-scheme (Walcek, 2000). The vertical diffusive transport (turbulent and molecular) is
calculated with the use of the implicit Thomas algorithm (Morton and Mayers, 1994). The model grid
includes 118 pressure-head levels (0–135 km), 16 latitudinal and 32 longitudinal levels. The chemical
module (see Table 1) comprises 25 constituents (O, O($^1$D), $O_3$, H, OH, $HO_2$, $H_2O_2$, $H_2O$, $H_2$, N, NO, $NO_2$,
$NO_3$, $N_2O$, $CH_4$, $CH_2$, $CH_3$, $CH_3O_2$, $CH_3O$, $CH_2O$, CHO, CO, $CO_2$, $O_2$, $N_2$), 54 chemical reactions



between them, and 15 photo-dissociation reactions. The model utilizes the pre-calculated dissociation
rates (Kremp et al., 1999) and their dependence on the altitude and solar zenith angle.

The model was used to calculate a one-year global evolution of the above mentioned trace gases.

For removing the transition regions corresponding to sunset and sunrise, we took into account the local
time when the solar zenith angle > 95°. As a result, we find the spatiotemporal series of the $OH/OH^{eq}$
and $HO_2/HO_2{}^{eq}$ ratios. Here, $OH$ and $HO_2$ are the local nighttime values of hydroxyl and hydroperoxy
radicals calculated by the model, $OH^{eq}$ and $HO_2{}^{eq}$ are their local equilibrium values corresponding to the
instantaneous balance between production and loss terms, respectively. Therefore, for determination of
each local value of $OH^{eq}$ and $HO_2{}^{eq}$, we used the local values of the parameters (temperature, $O_2$, and
$N_2$) and the concentrations of other trace gases determining local chemical sources and sinks of $OH$ and
$HO_2$. Then, the $OH/OH^{eq}$ and $HO_2/HO_2{}^{eq}$ series were averaged over the zonal coordinate and time
during each month and were presented as height-latitude maps, depending on the month. Each map
contains lines marking the boundaries of the equilibrium areas, where the following conditions are
satisfied:
$$\begin{cases} |< OH/OH^{eq} > -1| \le 0.1 \\ \quad \sigma_{OH/OH^{eq}} \le 0.1 \end{cases}, \begin{cases} |< HO_2/HO_2{}^{eq} > -1| \le 0.1 \\ \quad \sigma_{HO_2/HO_2{}^{eq}} \le 0.1 \end{cases}, \tag{1}$$

where the angle brackets are used to denote the values averaged in time and space, $\sigma_{OH/OH^{eq}}$ and
$\sigma_{HO_2/HO_2{}^{eq}}$ are standard deviations of the $OH/OH^{eq}$ and $HO_2/HO_2{}^{eq}$ ratios from 1, respectively.

Then, we plotted spatiotemporal maps showing the relative contribution of each reaction to a

summarized source or sink at all altitudes and latitudes. These maps helped us to identify the main
sources and sinks describing the chemical equilibrium of nighttime OH and $HO_2$ in the equilibrium areas
to an accuracy of better than a few percent.

At final stage, we obtained and verified the analytical criteria of OH and $HO_2$ nighttime chemical

equilibria according to Kulikov et al. (2023a). The paper considered the poorly chemical evolution of a
certain trace gas $n$. It was shown strictly mathematically, that the local values of $n$ and its equilibrium
concentration $n^{eq}$ are close to each other ($n(t) \approx n^{eq}(t)$), when $\tau_n \ll \tau_{n^{eq}}$, where $\tau_n$ is the $n$ lifetime
and $\tau_{n^{eq}}$ is the local time scale of $n^{eq}$:
$$\tau_{n^{eq}} \equiv \frac{n^{eq}}{|dn^{eq}/dt|}. \tag{2}$$
The expression for $\tau_n$ is found from total sink of $n$. The expression for $\tau_{n^{eq}}$ is derived with the use of
differential equations describing chemical evolution of other reacting components which determine the
expression for $n^{eq}$. Kulikov et al. (2023a) also showed the criterion



$\tau_n/\tau_n{}^{eq} \leq 0.1$ (3)
is sufficient, in order to the possible difference between $n$ and $n^{eq}$ to be no more than 0.1.

**3 Nighttime HO$_2$ and OH chemical equilibriums**

According to the Table 1, HO$_2$ chemical sources in nighttime are determined by the following
reactions:
H+O$_2$+M $\rightarrow$ HO$_2$+M (R20), OH+O$_3$ $\rightarrow$ O$_2$+HO$_2$ (R22), H$_2$O$_2$+OH $\rightarrow$ H$_2$O+HO$_2$ (R29), H$_2$O$_2$+O $\rightarrow$
OH+HO$_2$ (R19), CHO+O$_2$ $\rightarrow$ HO$_2$+CO (R40), CH$_3$O+O$_2$ $\rightarrow$ CH$_2$O+HO$_2$ (R37),
whereas chemical sinks of this component are as follows:
HO$_2$+O $\rightarrow$ OH+O$_2$ (R18), HO$_2$+O$_3$ $\rightarrow$ OH+2O$_2$ (R23), OH+HO$_2$ $\rightarrow$ H$_2$O+O$_2$ (R28), H+HO$_2$ $\rightarrow$ 2OH
(R14), H+HO$_2$ $\rightarrow$ H$_2$O+O (R15), H+HO$_2$ $\rightarrow$ H$_2$+O$_2$ (R16), NO+HO$_2$ $\rightarrow$ NO$_2$+OH (R50), HO$_2$+HO$_2$ $\rightarrow$
H$_2$O$_2$+O$_2$ (R30), HO$_2$+HO$_2$+M $\rightarrow$ H$_2$O$_2$+O$_2$+M (R31).
Thus, HO$_2$ local equilibrium concentration is described by the following equation:
$$HO_2{}^{eq} = \frac{k_{20} \cdot H \cdot M \cdot O_2 + k_{22} \cdot OH \cdot O_3 + k_{29} \cdot H_2O_2 \cdot OH + k_{19} \cdot H_2O_2 \cdot O + k_{40} \cdot CHO \cdot O_2 + k_{37} \cdot CH_3O \cdot O_2}{k_{18} \cdot O + k_{23} \cdot O_3 + k_{28} \cdot OH + (k_{14} + k_{15} + k_{16}) \cdot H + k_{50} \cdot NO + 2 \cdot (k_{30} + k_{31} \cdot M) \cdot HO_2}$$ (4)
The Figure 1 plots height-latitude cross sections for the $< HO_2/HO_2{}^{eq} >$ ratio for each month. The
dashed area corresponds to $\chi<95°$. The white area represents the ratio outside the [0.5, 1.5] interval. The
black solid lines mark the borders of equilibrium areas, where, according to (1), local values of HO$_2$ are
close to their equilibrium values with a possible bias of less than 10%. At low and middle latitudes, one
can see the present of the main equilibrium area, which extends from the top of the analyzed altitude
range to the lower boundary. The height of this equilibrium border, $z_{HO_2{}^{eq}}$, depends on the season and
latitude and varies in the interval between 73 and 85 km. It is the highest and the lowest during the
summer and winter respectively at the middle latitudes. Near equator, $z_{HO_2{}^{eq}}$ demonstrates the weakest
annual variations and varies in the range of 81-83 km. There are local areas below the upper longest black
line, but they are small and irregular and can be omitted from our consideration. Note only that the maps
in many months show the existence of equilibrium near 50 km, which can be assumed to be the beginning
of the main equilibrium area in the stratosphere. At high latitudes, there is the main equilibrium area as at
low and middle latitudes, but this area above 70-75° of latitude can extend down to 50 km with small
exceptions.
In accordance to the Table 1, OH chemical sources are determined by the following reactions:



H+O$_3$ → OH+O$_2$ (R21), HO$_2$+O → OH+O$_2$ (R18), HO$_2$+O$_3$ → OH+2O$_2$ (R23), H+HO$_2$ → 2OH (R14),
NO+HO$_2$ → NO$_2$+OH (R50), H$_2$O$_2$+O → OH+HO$_2$ (R19), H +NO$_2$ → OH+NO (R51), O($^1$D)+H$_2$O →
2OH (R7), O($^1$D)+H$_2$ → H+OH (R8), CH$_4$+O($^1$D) → CH$_3$+OH (R9),
whereas chemical sinks of this component are as follows:
OH+O → H+O$_2$ (R17), OH+O$_3$ → O$_2$+HO$_2$ (R22), OH+HO$_2$ → H$_2$O+O$_2$ (R28), OH+OH → H$_2$O+O
(R26), OH+OH+M → H$_2$O$_2$+M (R27), H+OH+N$_2$ → H$_2$O+N$_2$ (R24), H$_2$O$_2$+OH → H$_2$O+HO$_2$ (R29),
OH+CO → H+CO$_2$ (R32), CH$_4$+OH → CH$_3$+H$_2$O (R33), OH+H$_2$ → H$_2$O+H (R25), N+OH → NO+H
(R49).
Thus, OH local equilibrium concentration is described by the following equation:
$OH^{eq} = (k_{21} \cdot H \cdot O_3 + k_{18} \cdot O \cdot HO_2 + k_{23} \cdot HO_2 \cdot O_3 + 2 \cdot k_{14} \cdot H \cdot HO_2 + k_{50} \cdot HO_2 \cdot NO + k_{19} \cdot$
$H_2O_2 \cdot O + k_{24} \cdot H \cdot N_2 + k_{51} \cdot NO_2 \cdot H + 2 \cdot k_7 \cdot O(^1D) \cdot H_2O + k_8 \cdot O(^1D) \cdot H_2 + k_9 \cdot O(^1D) \cdot$
$CH_4)/(k_{17} \cdot O + k_{22} \cdot O_3 + k_{28} \cdot HO_2 + 2 \cdot (k_{26} + k_{27} \cdot M) \cdot OH + k_{29} \cdot H_2O_2 + k_{32} \cdot CO + k_{33} \cdot CH_4 +$
$k_{25} \cdot H_2 + k_{25} \cdot N)$                                                                  (5)

Figure 2 shows height-latitude cross sections for the $< OH/OH^{eq} >$ ratio for each month. In this

case, the equilibrium covers up to 70-80% of the presented ranges of heights and latitudes, so that the
black solid lines mark the external borders of non-equilibrium areas. In March and September, this area is
almost symmetrical to the equator. In April-August, it is shifted towards the northern hemisphere. In
October-February, this area is higher in the southern hemisphere. In all months, it is below 85-86 km. In
the polar regions, there are latitudinal ranges where OH is close to equilibrium throughout the entire range
of heights.

**4 The main reactions determining HO$_2$ and OH equilibriums**

The Figure 3 shows height-latitude contour maps showing the relative contribution of a certain

reaction to the total source of HO$_2$ in different months taken for example. To increase the information
content of the panels, the altitude range is cut off everywhere to 100 km, since there are no significant
changes above. Note, first, that reaction H+O$_2$+M → HO$_2$+M determines a major (up to 95% and more)
contribution in the main equilibrium area almost everywhere except for the polar regions above 70-75° of
latitude and below 75-80 km, where the reactions OH+O$_3$ → O$_2$+HO$_2$ and H$_2$O$_2$+OH → H$_2$O+HO$_2$
become important and should be taken into account. Second, other reactions (H$_2$O$_2$+O → OH+HO$_2$,
CHO+O$_2$ → HO$_2$+CO, CH$_3$O+O$_2$ → CH$_2$O+HO$_2$) together contribute less than 2-3% to the total source
of HO$_2$ in the main equilibrium area and may be omitted.



The Figure 4 presents height-latitude contour maps showing the relative contribution of a certain
reaction to the total sink of $HO_2$ in the same months as in Figure 3. Firstly, it should be noted that reaction
$HO_2+O \rightarrow OH+O_2$ determines a major (up to 95% and more) contribution in the main equilibrium area
almost everywhere except for the same small polar areas, as in the considered case with the sources,
where the reactions $HO_2+O_3 \rightarrow OH+2O_2$ and $NO+HO_2 \rightarrow NO_2+OH$ are important and should be taken
into account. Secondly, the reactions $OH+HO_2 \rightarrow H_2O+O_2$, $H+HO_2 \rightarrow 2OH$, $H+HO_2 \rightarrow H_2O+O$, and
$H+HO_2 \rightarrow H_2+O_2$ give together contribute up to 10-15% of the total source near the boundary of the main
equilibrium area. Thirdly, the remaining reactions ($HO_2+HO_2 \rightarrow H_2O_2+O_2$, $HO_2+HO_2+M \rightarrow$
$H_2O_2+O_2+M$) are not important in the main equilibrium area and can be omitted.
Therefore, the expression for $HO_2$ local equilibrium concentration can be simplified as follows:
$$HO_2{}^{eq} = \frac{k_{20} \cdot H \cdot M \cdot O_2 + k_{22} \cdot OH \cdot O_3 + k_{29} \cdot H_2O_2 \cdot OH}{k_{18} \cdot O + k_{23} \cdot O_3 + k_{28} \cdot OH + (k_{14}+k_{15}+k_{16}) \cdot H + k_{50} \cdot NO}$$    (6)
Figures 5-6 show height-latitude contour maps showing the relative contribution of a certain
reaction to the total source of OH in the same months as in Figure 3 taken for example. As in the previous
case, the altitude range is cut off at 100 km, because only the panels for the reactions $H+O_3 \rightarrow OH+O_2$
and $HO_2+O \rightarrow OH+O_2$ consist of interesting variations at the 100-130 km altitudes. Note that these
reactions are the main OH sources in the upper part of the presented distributions down to 70-75 km,
where they jointly provide up to 95% contribution in equilibrium concentration. Also, the reaction
$HO_2+O_3 \rightarrow OH+2O_2$ is major in the lower part of the presented distributions from 50 to 60-70 km,
depending on the month. The reaction $NO+HO_2 \rightarrow NO_2+OH$ is important around non-equilibrium areas
of OH and should be taken into account, whereas the reaction $H+NO_2 \rightarrow OH+NO$ is important in
compact altitude-latitude areas near the poles, the reaction $H+HO_2 \rightarrow 2OH$ gives up to 10-15%
contribution in small areas near the equilibrium boundary. Other reactions ($O(^1D)+H_2O \rightarrow 2OH$,
$O(^1D)+H_2 \rightarrow H+OH$, $CH_4+O(^1D) \rightarrow CH_3+OH$, $H_2O_2+O \rightarrow OH+HO_2$) together contribute less than 2-3%
of the total source of OH in the main equilibrium area and can be omitted.
Figures 7-8 present height-latitude contour maps showing the relative contribution of a certain
reaction to the total sink of OH. First, note that the reaction $OH+O \rightarrow H+O_2$ is the main OH sink in the
upper part of the presented distributions down to 70-80 km, depending on the month, where it provides up
to 95% of the equilibrium concentration. The reactions $OH+CO \rightarrow H+CO_2$ and $OH+O_3 \rightarrow O_2+HO_2$ are
major in the lover part of the presented distributions from 50 to 70-80 km, depending on the month. The
reaction $OH+HO_2 \rightarrow H_2O+O_2$ is remarkable around non-equilibrium areas of OH, whereas the reaction
$H_2O_2+OH \rightarrow H_2O+HO_2$ is important in the compact altitude-latitude area near the poles. Other reactions
($OH+OH \rightarrow H_2O+O$, $OH+H_2 \rightarrow H_2O+H$, $N+OH \rightarrow NO+H$, $CH_4+OH \rightarrow CH_3+H_2O$, $H+OH+N_2 \rightarrow$





$H_2O+N_2$, $OH+OH+M \rightarrow H_2O_2+M$) together contribute less than 2-3% to the total source of OH in the
main equilibrium area and can be omitted.
Therefore, the expression for OH local equilibrium concentration can be can be simplified as
follows:
$$OH^{eq} = \frac{k_{21} \cdot H \cdot O_3 + k_{18} \cdot O \cdot HO_2 + k_{23} \cdot HO_2 \cdot O_3 + 2 \cdot k_{14} \cdot H \cdot HO_2 + k_{24} \cdot H \cdot N_2 + k_{50} \cdot HO_2 \cdot NO + k_{51} \cdot NO_2 \cdot H}{k_{17} \cdot O + k_{22} \cdot O_3 + k_{28} \cdot HO_2 + k_{29} \cdot H_2O_2 + k_{32} \cdot CO} \qquad (7)$$

## 5 Shortened equilibrium conditions of HO$_2$ and OH in the upper mesosphere and lower thermosphere

**5 Shortened equilibrium conditions of HO$_2$ and OH in the upper mesosphere and lower**
**thermosphere**
The above analysis revealed that the reactions describing the equilibrium conditions (6-7) in the
lower and middle mesosphere are mainly different from those in the upper mesosphere and lower
thermosphere. This means that the task of applying these conditions can be divided into two parts
depending on the selected altitude range. At the upper mesosphere and lower thermosphere altitudes, we
can consider only the $HO_x - O_x$ chemistry, excluding the reactions with participation of $H_2O_2$, N, NO,
$NO_2$, and CO. In addition, we can omit the reactions $HO_2+O_3 \rightarrow OH+2O_2$, $OH+O_3 \rightarrow O_2+HO_2$, and
$OH+HO_2 \rightarrow H_2O+O_2$ due to their insignificance here. As the result, the shortened equilibrium conditions
of HO$_2$ and OH for this altitude range are as follows:
$$HO_{2_{sh}}^{eq} = \frac{k_{20} \cdot H \cdot M \cdot O_2}{k_{18} \cdot O + (k_{14}+k_{15}+k_{16}) \cdot H}, \qquad (8)$$
$$OH_{sh}^{eq} = \frac{k_{21} \cdot H \cdot O_3 + k_{18} \cdot O \cdot HO_2 + 2 \cdot k_{14} \cdot H \cdot HO_2}{k_{17} \cdot O} \qquad (9)$$
The Figure 9 shows height-latitude cross sections for the $< HO_2/HO_{2_{sh}}^{eq} >$ ratio for each month. In
each panel, the upper longest black line marks the lower border of the main equilibrium area, where,
according to (1), $HO_2 \approx HO_{2_{sh}}^{eq}$ with possible bias of less than 10%. As in the case of Figure 1, this area
extends from the top of the analyzed altitude range. There are also very small equilibrium areas below,
which can be omitted from our consideration. The height of the lower border of the main equilibrium
area, $z_{HO_{2_{sh}}^{eq}}$, depends essentially on the season and latitude. Comparing with Figure 1, one can see, it
repeats well many features of $z_{HO_{2_{sh}}^{eq}}$ at low and middle latitudes. In particular, $z_{HO_{2_{sh}}^{eq}}$ varies in the
interval between 73 and 85 km, as in the case of $z_{HO_2^{eq}}$. In the middle latitudes, $z_{HO_{2_{sh}}^{eq}}$ in summer is
several km higher than in winter. Near equator, $z_{HO_{2_{sh}}^{eq}}$ demonstrates the weakest annual variations and
varies in the range of 81-83 km. So, one can conclude that the exclusion of a number of reactions does not
lead to significant changes in the space-time distributions of the HO$_2$ equilibrium.



The Figure 10 plots height-latitude cross sections for the $< OH/OH_{sh}^{eq} >$ ratio for each month. As
in the previous case, this is the lower border of the equilibrium area, where, according to (1), $OH \approx OH_{sh}^{eq}$
with good precision. The dependence of the border height, $z_{OH_{sh}^{eq}}$, on the season and latitude repeats
mainly $z_{HO_{2sh}^{eq}}$. In particular, $z_{OH_{sh}^{eq}}$ varies in the interval between 73 and 85 km. At middle latitudes,
$z_{OH_{sh}^{eq}}$ in summer is several km higher than in winter. Near the equator, $z_{OH_{sh}^{eq}}$ also demonstrates the
weakest annual variations and varies in the range of 81-83 km. Nevertheless, in some cases, the OH
equilibrium border lies slightly higher than the $HO_2$ border. In particular, it can be seen in April-August
above 50ºS, which can be explained by the difference between $HO_2$ and OH lifetimes ($\tau_{HO_2} < \tau_{OH}$),
mainly, due to $k_{18} > k_{17}$. Comparing with Figure 2, one can see that the exclusion of the mentioned
reactions from consideration results in the absence of the OH equilibrium areas at the low and middle
mesosphere altitudes, as expected.

**6 The criterions for $HO_2$ and OH equilibrium validity in the upper mesosphere and lower**
**thermosphere**
Let determine $HO_2$ and OH lifetimes and the local time scales of $HO_{2sh}^{eq}$ and $OH_{sh}^{eq}$, according to
Section 2.
From (8), $HO_2$ lifetime and the local time scales of $HO_{2sh}^{eq}$ are as follows:
$\tau_{HO_2} = \frac{1}{k_{18} \cdot O + (k_{14} + k_{15} + k_{16}) \cdot H}$,                                         (10)
$\tau_{HO_{2sh}^{eq}} = \frac{HO_{2sh}^{eq}}{|dHO_{2sh}^{eq}/dt|}$.                                         (11)
Let find the expression for $dHO_{2sh}^{eq}/dt$:
$\frac{dHO_{2sh}^{eq}}{dt} = \frac{k_{18} \cdot k_{20} \cdot M \cdot O_2 \cdot \frac{d}{dt}(\frac{H}{O}) \cdot O^2}{(k_{18} \cdot O + (k_{14} + k_{15} + k_{16}) \cdot H)^2} = -\frac{k_{18} \cdot k_{20} \cdot M \cdot O_2 \cdot \frac{d}{dt}(\frac{O}{H}) \cdot H^2}{(k_{18} \cdot O + (k_{14} + k_{15} + k_{16}) \cdot H)^2}$.                                         (12)
Kulikov et al. (2023a) analyzed analytically the local nighttime evolution of O and H within the
framework of pure $HO_x$ – $O_x$ chemistry and found the expression for $\frac{d}{dt}\left(\frac{O}{H}\right)$:
$\frac{d}{dt}\left(\frac{O}{H}\right) = -2 \cdot k_{20} \cdot M \cdot O_2 \cdot \left(1 - \frac{k_{15} + k_{16}}{k_{18}}\right) - k_{21} \cdot O_3 - k_{12} \cdot M \cdot O_2 \cdot \frac{O}{H}$.                                         (13)
Thus, the expression (12) can be rewritten in following form:
$\frac{dHO_{2sh}^{eq}}{dt} = \frac{k_{18} \cdot k_{20} \cdot M \cdot O_2 \cdot H^2 \cdot \left(2 \cdot k_{20} \cdot M \cdot O_2 \cdot \left(1 - \frac{k_{15} + k_{16}}{k_{18}}\right) + k_{21} \cdot O_3 + k_{12} \cdot M \cdot O_2 \cdot \frac{O}{H}\right)}{(k_{18} \cdot O + (k_{14} + k_{15} + k_{16}) \cdot H)^2}$.                                         (14)



By combining (8), (11), and (14) we obtain the expression for the local time scales of $HO_{2_{sh}}^{eq}$:
$\tau_{HO_{2_{sh}}^{eq}} = \frac{(k_{18}\cdot O+(k_{14}+k_{15}+k_{16})\cdot H)}{k_{18}\cdot H\cdot\left(2\cdot k_{20}\cdot M\cdot O_2\cdot\left(1-\frac{k_{15}+k_{16}}{k_{18}}\right)+k_{21}\cdot O_3+k_{12}\cdot M\cdot O_2\cdot\frac{O}{H}\right)}$      (15)
Thus, taking into account (3), (10) and (15), the criterion for HO$_2$ equilibrium validity is written in the
form:
$Crit_{HO_2} = \frac{\tau_{HO_2}}{\tau_{\tau_{HO_{2_{sh}}^{eq}}}} = \frac{k_{18}\cdot H\cdot\left(2\cdot k_{20}\cdot M\cdot O_2\cdot\left(1-\frac{k_{15}+k_{16}}{k_{18}}\right)+k_{21}\cdot O_3+k_{12}\cdot M\cdot O_2\frac{O}{H}\right)}{(k_{18}\cdot O+(k_{14}+k_{15}+k_{16})\cdot H)^2} \leq 0.1.$      (16)
We calculated $Crit_{HO_2}$ using the global 3D chemical transport model and included the zonally and
monthly averaged lines $< Crit_{HO_2} >= 0.1$ in Figure 9 (see magenta lines). One can see that, depending
on the month, each red line well reproduces the lower border of the main OH equilibrium area and repeats
almost all its features and variations. Note that, in zero approximation, the criterion (16) can be simplified
as
$Crit_{HO_2} \approx \left(2\cdot k_{20}\cdot M\cdot O_2\cdot\left(1-\frac{k_{15}+k_{16}}{k_{18}}\right)+k_{21}\cdot O_3+k_{12}\cdot M\cdot O_2\cdot\frac{O}{H}\right)\cdot\frac{H}{k_{18}\cdot O^2}\cdot\leq 0.1.$      (17)
From (9), OH lifetime and the local time scales of $OH_{sh}^{eq}$ are as follows:
$\tau_{OH} = \frac{1}{k_{17}\cdot O},$      (18)
$\tau_{OH_{sh}^{eq}} = \frac{OH_{sh}^{eq}}{|dOH_{sh}^{eq}/dt|}.$      (19)
Before determining the expression for $dOH_{sh}^{eq}/dt$, first of all, one should to keep in mind that the
expression (9) depends on the HO$_2$ concentration. Above mentioned, that near and above the OH
equilibrium border, HO$_2$ is in equilibrium ($HO_2 \approx HO_{2_{sh}}^{eq}$) and we can use expression (8). In view of
$k_{18}\cdot O \gg (k_{14}+k_{15}+k_{16})\cdot H$,
$HO_{2_{sh}}^{eq} \approx \frac{k_{20}\cdot H\cdot M\cdot O_2}{k_{18}\cdot O}(1-\frac{(k_{14}+k_{15}+k_{16})\cdot H}{k_{18}\cdot O}).$      (20)
The substitution of (20) into (9) yields:
$OH_{sh}^{eq} = k_{20}\cdot H\cdot M\cdot O_2\cdot\frac{\left(1+\frac{2\cdot k_{14}\cdot H}{k_{18}\cdot O}\right)\cdot\left(1-\frac{(k_{14}+k_{15}+k_{16})\cdot H}{k_{18}\cdot O}\right)}{k_{17}\cdot O}+\frac{k_{21}\cdot H\cdot O_3}{k_{17}\cdot O} \approx \frac{k_{20}\cdot H\cdot M\cdot O_2}{k_{17}\cdot O}\cdot\left(1+\frac{(k_{14}-k_{15}-k_{16})\cdot H}{k_{18}\cdot O}\right)+$
$\frac{k_{21}\cdot H\cdot O_3}{k_{17}\cdot O}$      (21)
Thus, the expression for $dOH_{sh}^{eq}/dt$ is:
$\frac{dOH_{sh}^{eq}}{dt} = \frac{d}{dt}\left(\frac{H}{O}\right)\cdot\left(\frac{k_{20}\cdot M\cdot O_2}{k_{17}}\cdot\left(1+\frac{2\cdot(k_{14}-k_{15}-k_{16})\cdot H}{k_{18}\cdot O}\right)+\frac{k_{21}\cdot O_3}{k_{17}}\right)+\frac{k_{21}\cdot H}{k_{17}\cdot O}\frac{dO_3}{dt}.$      (22)



Taking into account (13) and the differential equation for $O_3$ time evolution:
$\frac{dO_3}{dt} = k_{12} \cdot M \cdot O_2 \cdot O - k_{21} \cdot H \cdot O_3,$
the expression (21) can be rewritten in following form:
$\frac{dOH_{sh}^{eq}}{dt} =$
$(2 \cdot k_{20} \cdot M \cdot O_2 \cdot \left(1 - \frac{k_{15}+k_{16}}{k_{18}}\right) + k_{21} \cdot O_3 + k_{12} \cdot M \cdot O_2 \cdot \frac{O}{H}) \cdot \frac{H^2}{O^2} \left(\frac{k_{20} \cdot M \cdot O_2}{k_{17}} \cdot \left(1 + \frac{2 \cdot (k_{14} - k_{15} - k_{16}) \cdot H}{k_{18} \cdot O}\right) +$
$k21 \cdot O3k17 + k21 \cdot H \cdot (k12 \cdot M \cdot O2 \cdot O - k21 \cdot H \cdot O3)k17 \cdot O.$    (23)
Thus, by combining (3), (18), (19), (21), and (23) we obtain the expression for the criterion for OH
equilibrium validity:
$Crit_{OH} = \frac{\tau_{OH}}{\tau_{\tau_{OH_{sh}^{eq}}}} = \frac{1}{k_{17} \cdot O} \cdot \left(\left(2 \cdot k_{20} \cdot M \cdot O_2 \cdot \left(1 - \frac{k_{15}+k_{16}}{k_{18}}\right) + k_{21} \cdot O_3 + k_{12} \cdot M \cdot O_2 \cdot \frac{O}{H}\right) \cdot \frac{H}{O} \cdot$
$k20 \cdot M \cdot O2 \cdot 1 + 2 \cdot k14 - k15 - k16 \cdot Hk18 \cdot O + k21 \cdot O3 + k21 \cdot k12 \cdot M \cdot O2 \cdot O - k21 \cdot H \cdot O3/(k20 \cdot M \cdot O2 \cdot (1 + k14 - k15 - k16 \cdot Hk18 \cdot O) + k21 \cdot$
$O_3) \leq 0.1.$    (24)
We calculated $Crit_{OH}$ using the global 3D chemical transport model and included the zonally and
monthly averaged lines $< Crit_{OH} >= 0.1$ in Figure 10 (see magenta lines). One can see that, depending
on the month, red line almost everywhere reproduces the lower border of the OH equilibrium area and
repeats mainly its features and variations. Nevertheless, there are a few (by latitude) narrow areas (in
April-August near 70ºS and in October-December near 70ºN) where the criterion gives a few km lower
position of the OH equilibrium boundary, that is going to be discussed below. Note our numerical
analysis shows that, in zero approximation, the criterion (24) can be simplified as:
$Crit_{OH} \approx \left(2 \cdot k_{20} \cdot M \cdot O_2 \cdot \left(1 - \frac{k_{15}+k_{16}}{k_{18}}\right) + k_{21} \cdot O_3 + k_{12} \cdot M \cdot O_2 \cdot \frac{O}{H}\right) \cdot \frac{H}{k_{17} \cdot O^2} \cdot \leq 0.1.$    (25)

**7 Discussion**

Let's discuss obtained results and their possible applications.

As noted, Figures 9-10 present an interesting peculiarity. At middle latitudes, summer $z_{HO_{2_{sh}}^{eq}}$ and

$z_{OH_{sh}^{eq}}$ are several km higher than winter ones. Recently (Kulikov et al., 2023b) such a feature was found
in the evolution of nighttime ozone chemical equilibrium boundary derived from SABER/TIMED data,
whichwas accompanied by the same variation of the transition zone dividing deep and weak



photochemical oscillations of O and H caused by the diurnal variations of solar radiation. Kulikov et al.
(2023b) analyzed this effect analytically and explained by the markedly lower values of the O and H
nighttime evolution times in summer than in winter by virtue, mainly, of the lower values of the $O/H$
ratio during the night, which, in turn, is determined by the daytime photochemistry. At middle, the ozone
boundary varies within 4-5 km interval above 80 km, whereas the range of OH and $HO_2$ boundaries
variations is 72-85 km (see Figures 9-10). In the case of ozone, its criterion for equilibrium validity (see
(5) in Kulikov et al. (2023b)) is as follows:

$$Crit_{O_3} = 2\frac{k_{12} \cdot O_2 \cdot M}{k_{21}}\left(k_{20} \cdot M \cdot O_2 \cdot \left(1 - \frac{k_{15}+k_{16}}{k_{18}}\right) + k_{21} \cdot O_3\right) \cdot \frac{1}{k_{21} \cdot H \cdot O_3} \leq 0.1. \tag{26}$$

At $O_3 \approx O_3^{eq} = \frac{k_{12} \cdot M \cdot O_2 \cdot O}{k_{21} \cdot H}$, one can see that $Crit_{O_3} \sim \frac{1}{O}$. It is follows from simplified expressions (17) and
(25) that $Crit_{HO_2}$ and $Crit_{OH}$ are proportional to $\frac{H}{O^2}$. Such dependence leads to a stronger annual variation
of OH and $HO_2$ equilibrium boundaries than in the case of $O_3$.

As noted, there are a few narrow areas near 70ºS,N (Figure 10) where the criterion (25) does not
agree well with the OH equilibrium boundary. Our analysis revealed that the main reason is neglecting
the reaction OH+CO $\rightarrow$ H+$CO_2$ as the source of H in the corresponding differential equation of its
chemical balance. In order to improve the criterion, we revised the derivation of expression (17) for $\frac{d}{dt}\left(\frac{O}{H}\right)$
following to Kulikov et al. (2023a):

$$\frac{d}{dt}\left(\frac{O}{H}\right) = -2 \cdot k_{20} \cdot M \cdot O_2 \cdot \left(1 - \frac{k_{15}+k_{16}}{k_{18}}\right) - k_{21} \cdot O_3 - k_{12} \cdot M \cdot O_2 \cdot \frac{O}{H} - \frac{k_{32} \cdot CO}{k_{17} \cdot H} \cdot (k_{20} \cdot M \cdot O_2 \cdot (1 +$$
$$\frac{(k_{14}-k_{15}-k_{16}) \cdot H}{k_{18} \cdot O}) + k_{21} \cdot O_3)). \tag{27}$$

As the result, the corrected criterion for OH equilibrium validity is as follows:

$$Crit_{OH}{}^m = \frac{1}{k_{17} \cdot O} \cdot \left(\left(2 \cdot k_{20} \cdot M \cdot O_2 \cdot \left(1 - \frac{k_{15}+k_{16}}{k_{18}}\right) + k_{21} \cdot O_3 + k_{12} \cdot M \cdot O_2 \cdot \frac{O}{H} + \frac{k_{32} \cdot CO}{k_{17} \cdot H} \cdot (k_{20} \cdot M \cdot O_2 \cdot\right.\right.$$
$$(1+k14-k15-k16Hk18·O)+k21·O3))·HO·k20·M·O2·1+2·k14-k15-k16Hk18·O+k21·O3+k21·k12·M·O2·O-k21·H·O3/(k20·$$
$$M \cdot O_2 \cdot (1 + \frac{(k_{14}-k_{15}-k_{16}) \cdot H}{k_{18} \cdot O}) + k_{21} \cdot O_3) \leq 0.1. \tag{28}$$

We calculated this criterion using the global 3D chemical transport model and included the zonally and
monthly averaged lines $< Crit_{OH}{}^m > = 0.1$ on the OH equilibrium maps (see Figure 11). One can see
that including additional term actually eliminates the noted discrepancy between OH boundary and
criterion. But, the application of this criterion requires CO data.



As noted in Introduction, the conditions of nighttime OH and $HO_2$ equilibriums together with one
for $O_3$ equilibrium and their analytical criteria constitute the useful tool for to retrieval these components
or other characteristics (for example, O and H) from measured data. At the altitudes of upper mesosphere
– lower thermosphere, these conditions can be applied, for example, to MLS/Aura database (measured
characteristics: OH, $HO_2$, $O_3$, and CO), SMILES ($HO_2$ and $O_3$), SCIAMACHY ($O(^1S)$ green-line, $O_2$ A-
band, and OH Meinel band emissions), SABER/TIMED ($O_3$, OH Meinel band emissions at 2.0 μm (9→7
and 8→6 bands) and at 1.6 μm (5→3 and 4→2 bands)) and other, including to improve existing retrieval
approaches. In particular, Panka et al. (2021) proposed the method of simultaneous derivation of O and
OH at the levels v=0-9 from SABER data (volume emission rates at 2.0 and 1.6 μm) at 80-100 km, taking
into account the equilibrium condition for all states of OH. Such approach is valid for exited states due to
its very low lifetimes determined by radiative transitions and quenching with $O_2$, $N_2$, and O. In the case of
the OH ground state, its lifetimes determined the reaction $OH+O \rightarrow H+O_2$ only. It means that Panka et al.
(2021) used an equilibrium condition for total OH, which may be significantly disrupted above 80 km in
certain latitude ranges and seasons, as one can see from Figure 8. In order to check this assumption, we
processed the Panka et al. (2021) data (O, OH($v$=9), $O_2$, $N_2$, and temperature profiles at 80-100 km) for
2009 and calculated local profiles of $Crit_{OH}$ according to criterion (24). One can see this criterion
depends of O, H, and $O_3$. Thus, the $O_3$ data was taken from SABER data collocated (via the orbit number)
with the Panka et al. profiles in time and space. The H data was derived with the use of the equilibrium
equation for OH($v$=9):
$$a_7 \cdot O_3 \cdot H = OH(9) \cdot (a_1 + a_2 \cdot N_2 + a_3 \cdot O_2 + (a_4 + a_5 + a_6) \cdot O), \qquad (29)$$
where $a_{1-7}$ are the constant rates of processes OH(9) → OH($v$≤8) + h$v$, OH(9) +$N_2$ → OH(8) + $N_2$,
OH(9) +$O_2$ → OH($v$≤8) + $O_2$, OH(9) +$O(^3P)$ → OH($v$≤4) + $O(^1D)$, OH(9) +$O(^3P)$ → OH($v$≤8) + $O(^3P)$,
OH($v$) +$O(^3P)$ → H + $O_2$, and H + $O_3$ → $O_2$ + OH(9), respectively. The values of $a_{2-7}$ correspond to the
Panka et al. (2021) model (see Table 1 there), the Einstein coefficients for OH($v$=9) were taken from
Brooke et al. (2016). Due to the strong air-concentration dependence $Crit_{OH}$ decreases rapidly with the
height. From each $Crit_{OH}$ profile, we determined the local height position of the OH equilibrium
boundary ($z_{OH_{sh}^{eq}}$) according to the condition $Crit_{OH} = 0.1$. It was revealed that $Crit_{OH} < 0.1$ throughout
the entire altitude range for most profiles. The Figure 12 plots the found values of $z_{OH_{sh}^{eq}}$ above 80 km in
different months. One can see that, in accordance of the Panka et al. data, the local height position of the
OH equilibrium boundary can rise up to 87 km. Moreover, the Panka et al. method requires external data
about $HO_2$ since the reaction $HO_2+O$ → $OH+O_2$ become the important source for OH below 87 km
(Panka et al., 2021; see also Figure 5 in our paper). Note that the $HO_2$ equilibrium condition (8) depends
on H and O only and can be used within the general retrieval procedure of O, H, OH($v$=0-9), and $HO_2$,



taking into account the criteria (16) and (24). Detailed development of this retrieval method is outside of
this paper and should be carried out in a separate extended work.

**8 Conclusions**

The presented analysis shows that there are extended areas in mesosphere and lower thermosphere,

where nighttime $HO_2$ and OH are close to their local equilibrium concentrations determined mainly by the
reaction between $HO_x – O_x$ components themselves and with $H_2O_2$, N, NO, $NO_2$, and CO. In upper
mesosphere – lower thermosphere, the shortened expressions for their local equilibrium concentrations
are valid, including the $HO_x – O_x$ chemistry only. These conditions describes the $HO_2$ and OH
equilibrium from the top to the some lower borders, the altitude position of which vary in the interval
between 73 and 85 km and depends essentially on the season and latitude. We proposed analytical
criteria, which almost everywhere well reproduces the main features of these borders. The obtained
results allow extending the abilities of the Panka et al. (2021) method of retrieval of unmeasured
components from SABER data. The simultaneous application of OH and $HO_2$ equilibrium conditions to
the SABER data ($O_3$, volume emission rates at 2.0 and 1.6 μm) together with the criteria (16) and (24) to
control this equilibrium validity is going to retrieve all unknown $HO_x – O_x$ components (O, H, OH, and
$HO_2$), extending the altitude range of retrieval below 80 km and without external information.

**Data availability.** The Panka et al. data are obtained from the SABER website
(https://saber.gats-inc.com).

**Code availability**. Code is available upon request.

**Author contributions.** MK and MB carried out the data processing and analysis and wrote the
manuscript. AC, SD, and AM contributed to reviewing the article.

**Competing interests.** The authors declare that they have no conflict of interest.

**Acknowledgements.**





**Financial support.** This work was supported by the Russian Science Foundation under grant No. 22-12-
00064 (https://rscf.ru/project/22-12-00064/) and state assignment no. 0729-2020-0037.

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





**Table 1.** List of reactions included in 3-d chemical transport model with the corresponding reaction rates
taken from Burkholder et al. (2020).

| 1 | $O(^1D)+O_2 \rightarrow O+O_2$ | 24 | $H+OH+N_2 \rightarrow H_2O+N_2$ | 47 | $NO+O_3 \rightarrow NO_2+O_2$ |
|---|---|---|---|---|---|
| 2 | $O(^1D)+N_2 \rightarrow O+N_2$ | 25 | $OH+H_2 \rightarrow H_2O+H$ | 48 | $NO_2+O_3 \rightarrow NO_3+O_2$ |
| 3 | $O(^1D)+O_3 \rightarrow O_2+2O$ | 26 | $OH+OH \rightarrow H_2O+O$ | 49 | $N+OH \rightarrow NO+H$ |
| 4 | $O(^1D)+O_3 \rightarrow 2O_2$ | 27 | $OH+OH+M \rightarrow H_2O_2+M$ | 50 | $NO+HO_2 \rightarrow NO_2+OH$ |
| 5 | $O(^1D)+N_2O \rightarrow 2NO$ | 28 | $OH+HO_2 \rightarrow H_2O+O_2$ | 51 | $H +NO_2 \rightarrow OH+NO$ |
| 6 | $O(^1D)+N_2O \rightarrow N_2+O_2$ | 29 | $H_2O_2+OH \rightarrow H_2O+HO_2$ | 52 | $NO_3+NO \rightarrow 2NO_2$ |
| 7 | $O(^1D)+H_2O \rightarrow 2OH$ | 30 | $HO_2+HO_2 \rightarrow H_2O_2+O_2$ | 53 | $N+NO \rightarrow N_2+O$ |
| 8 | $O(^1D)+H_2 \rightarrow H+OH$ | 31 | $HO_2+HO_2+M \rightarrow H_2O_2+O_2+M$ | 54 | $N+NO_2 \rightarrow N_2O+O$ |
| 9 | $O(^1D)+CH_4 \rightarrow CH_3+OH$ | 32 | $OH+CO \rightarrow H+CO_2$ | 55 | $O_2+h\nu \rightarrow 2O$ |
| 10 | $O(^1D)+CH_4 \rightarrow H_2+ CH_2O$ | 33 | $CH_4+OH \rightarrow CH_3+H_2O$ | 56 | $O_2+h\nu \rightarrow O+O(^1D)$ |
| 11 | $O+O+M \rightarrow O_2+M$ | 34 | $CH_3+O_2 \rightarrow CH_3O_2$ | 57 | $O_3+h\nu \rightarrow O_2+O$ |
| 12 | $O+O_2+M \rightarrow O_3+M$ | 35 | $CH_3+O \rightarrow CH_2O+H$ | 58 | $O_3+h\nu \rightarrow O_2+O(^1D)$ |
| 13 | $O+O_3 \rightarrow O_2 +O_2$ | 36 | $CH_3O_2+NO \rightarrow CH_3O+NO_2$ | 59 | $N_2+h\nu \rightarrow 2N$ |
| 14 | $H+HO_2 \rightarrow 2OH$ | 37 | $CH_3O+O_2 \rightarrow CH_2O+HO_2$ | 60 | $NO+h\nu \rightarrow N+O$ |
| 15 | $H+HO_2 \rightarrow H_2O+O$ | 38 | $CH_2O \rightarrow H_2+CO$ | 61 | $NO_2+h\nu \rightarrow NO+O$ |
| 16 | $H+HO_2 \rightarrow H_2+O_2$ | 39 | $CH_2O \rightarrow H+CHO$ | 62 | $N_2O+h\nu \rightarrow N_2+O(^1D)$ |
| 17 | $OH+O \rightarrow H+O_2$ | 40 | $CHO+O_2 \rightarrow HO_2+CO$ | 63 | $N_2O+h\nu \rightarrow N+NO$ |
| 18 | $HO_2+O \rightarrow OH+O_2$ | 41 | $O_3+N \rightarrow NO+O_2$ | 64 | $NO_3+h\nu \rightarrow NO_2+O$ |
| 19 | $H_2O_2+O \rightarrow OH+HO_2$ | 42 | $NO_3+O \rightarrow NO_2+O_2$ | 65 | $H_2O+h\nu \rightarrow H+OH$ |
| 20 | $H+O_2+M \rightarrow HO_2+M$ | 43 | $O+NO+M \rightarrow NO_2+M$ | 66 | $H_2O_2+h\nu \rightarrow 2OH$ |
| 21 | $H+O_3 \rightarrow OH+O_2$ | 44 | $NO_2+O \rightarrow NO+O_2$ | 67 | $CH_4+h\nu \rightarrow CH_2+H_2$ |
| 22 | $OH+O_3 \rightarrow O_2+HO_2$ | 45 | $NO_2+O+M \rightarrow NO_3+M$ | 68 | $CH_4+h\nu \rightarrow CH+H_2+H$ |
| 23 | $HO_2+O_3 \rightarrow OH+2O_2$ | 46 | $N+O_2 \rightarrow NO+O$ | 69 | $CO_2+h\nu \rightarrow CO+O$ |








Figure 1. Nighttime mean and monthly averaged $HO_2/HO_2{}^{eq}$. Black line shows the border of $HO_2$

equilibrium according to condition (1).




Figure 2. Nighttime mean and monthly averaged $OH/OH^{eq}$. Black line shows the border of OH equilibrium according to condition (1).



Figure 3. Nighttime mean and monthly averaged the relative contribution of a certain reaction to the total source of $HO_2$ in equilibrium areas. White color points nonequilibrium areas of $HO_2$.





Figure 4. Nighttime mean and monthly averaged the relative contribution of a certain reaction to the total sink of $HO_2$ in equilibrium areas. White color points nonequilibrium areas of $HO_2$.



Figure 5. Nighttime mean and monthly averaged the relative contribution of a certain reaction to the total source of OH in equilibrium areas (first part). White color points nonequilibrium areas of OH.

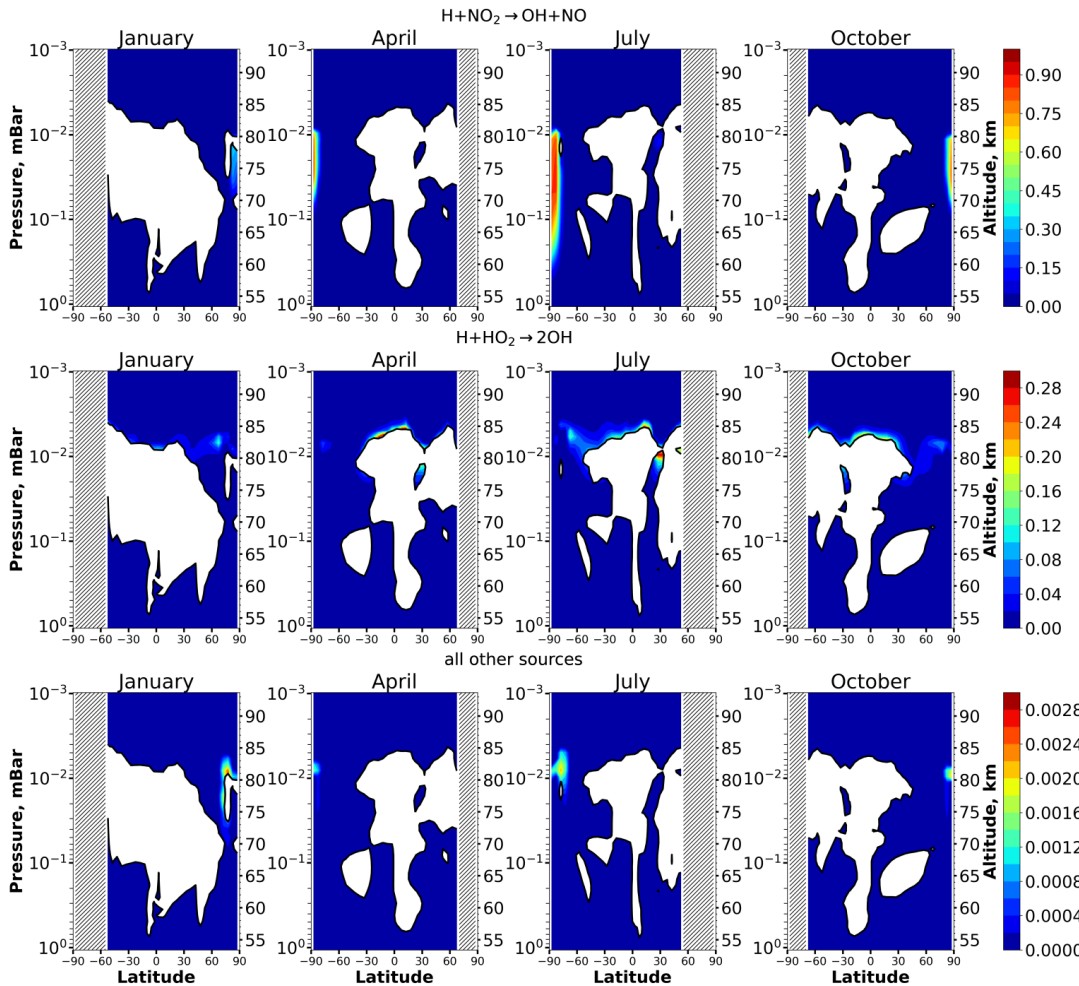

Figure 6. Nighttime mean and monthly averaged the relative contribution of a certain reaction to the total

source of OH in equilibrium areas (second part). White color points nonequilibrium areas of OH.





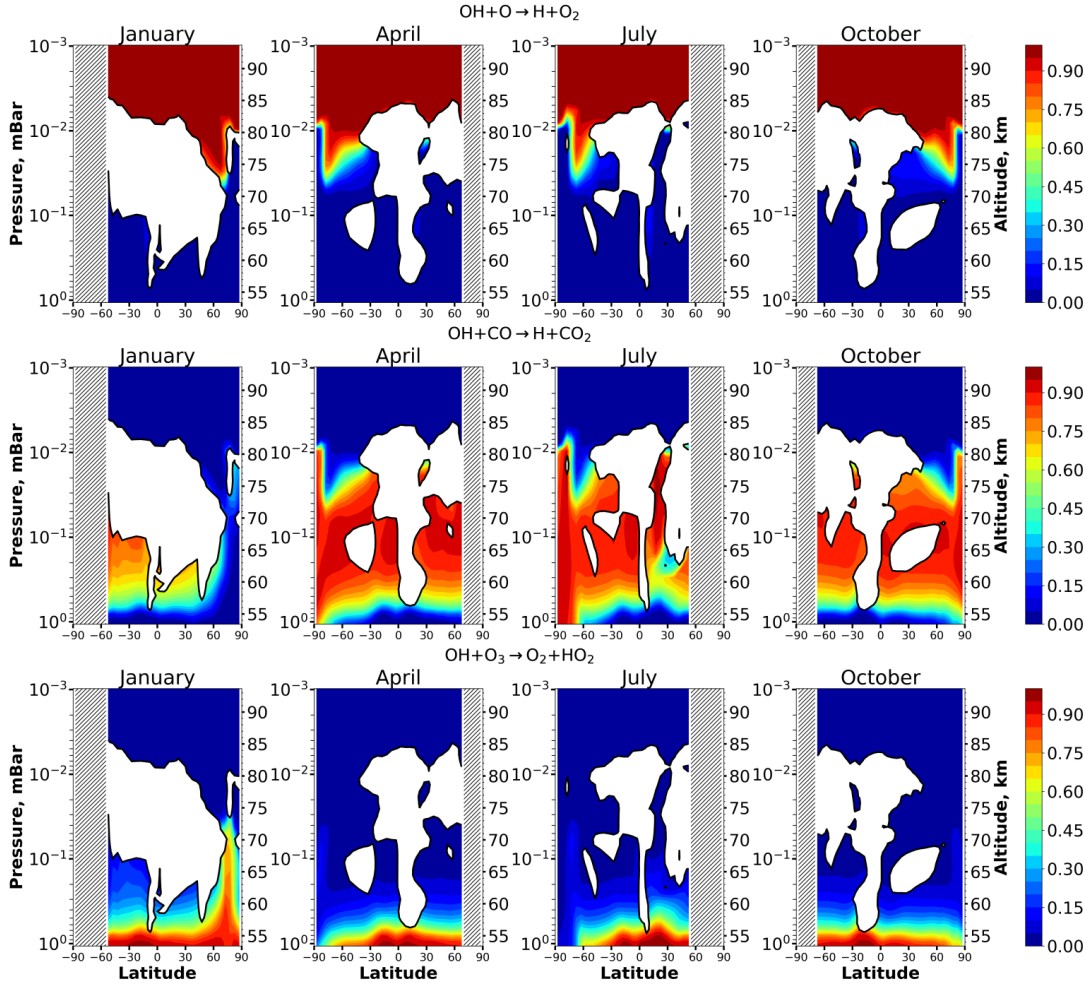

Figure 7. Nighttime mean and monthly averaged the relative contribution of a certain reaction to the total sink of OH in equilibrium areas (first part). White color points nonequilibrium areas of OH.



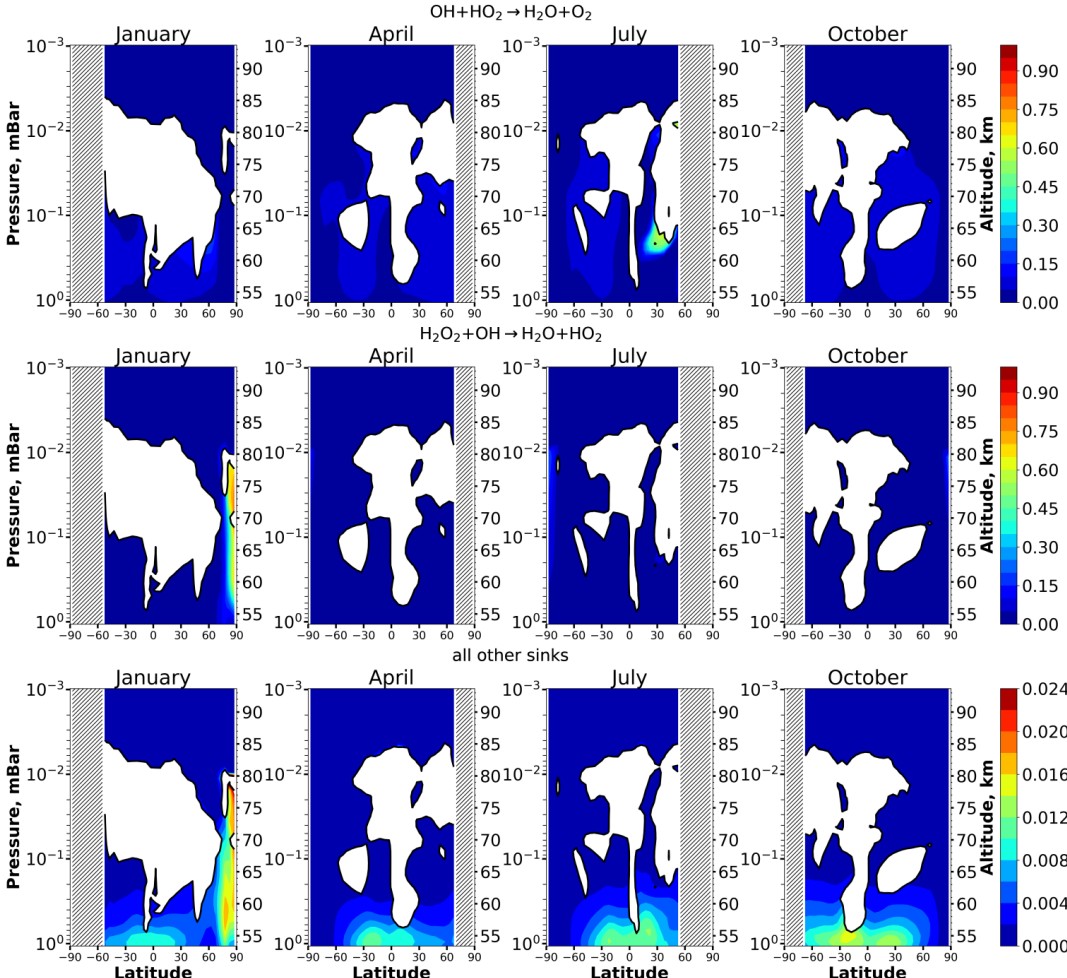


Figure 8. Nighttime mean and monthly averaged the relative contribution of a certain reaction to the total
source of OH in equilibrium areas (second part). White color points nonequilibrium areas of OH.



Figure 9. Nighttime mean and monthly averaged $HO_2/HO_{2_{sh}}^{eq}$. Black line shows the border of $HO_2$ equilibrium according to condition (1). Magenta line shows $<Crit_{HO_2}> = 0.1$.

Figure 10. Nighttime mean and monthly averaged $OH/OH_{sh}^{eq}$. Black line shows the border of OH equilibrium according to condition (1). Magenta line shows $< Crit_{OH} > = 0.1$.



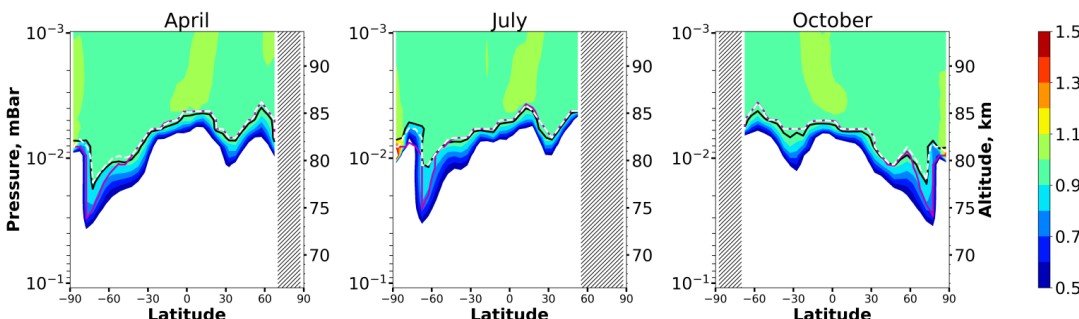

Figure 11. Nighttime mean and monthly averaged $OH/OH_{sh}^{eq}$. Black line shows the border of OH equilibrium according to condition (1). Magenta line shows $< Crit_{OH} >= 0.1$, dotted white line shows $< Crit_{OH}{}^{m} >= 0.1$.





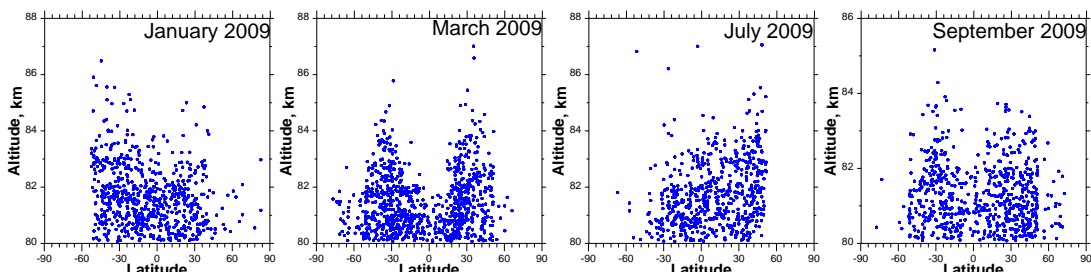

The Figure 12. Blue dots point the found values of $z_{OH_{sh}^{eq}}$ above 80 km derived from the Panka et al. data

in different months of 2009.