# Peer review of "Technical Note: Nighttime OH and HO2 chemical equilibria in the mesosphere – lower thermosphere"

_EGUsphere, 2024_

## Author Comment (AC1)

Dear Editor,

We would like to say many thanks the Referee for taking the time to review our manuscript and providing valuable recommendations. Their constructive criticism made the work clearer and more precise. We took into account all the remarks of Referee and, to the best of our ability, implemented the corresponding changes in the manuscript.

In the following, we address the comments point by point and show how the manuscript has been changed according to the comments. Below we use a certain color notation: comments by Referee are in red, our responses are in black, and the changes in the manuscript are in blue (placed inside the quotation marks).

**Response to the comments on the paper by Referee 1**

General Comments

…….

I am not clear on what significant new knowledge is gained. Some of the conclusions seem unsubstantiated and even unrealistic, especially the main conclusion that "the simultaneous application of OH and HO2 equilibrium to the SABER data (O3, volume emission rates at 2.0 and 1.6 um) together with the criteria (16) and (24) to control this equilibrium validity is going to retrieve all unknown HOx – Ox components (O, H, OH, and HO2), extending the altitude range of retrieval below 80 km and without external information."

In the revised manuscript, we give the detailed explanation of the noted conclusion (see Discussion):

«The results of our paper allow modifying the Panka et al. method to extend its capabilities. The simplest development of this method seems to be the following. First of all, note that the $HO_2$ equilibrium condition (9) depends on H and O only and can be used within the self-consistent retrieval procedure, considering the following system of equations:

$$OH(v=1-9) = \frac{k_{12} \cdot H \cdot O_3 \cdot M \cdot f(v) + \sum_{v'>v}(a_1(v',v) + a_2(v',v) \cdot O_2 + a_3(v',v) \cdot N_2 + (a_4(v',v) + a_5(v',v)) \cdot O) \cdot OH(v')}{a_6(v)) \cdot O + \sum_{v>v}'(a_1(v,v') + a_2(v,v') \cdot O_2 + a_3(v,v') \cdot N_2 + (a_4(v,v') + a_5(v,v')) \cdot O)},$$

$$OH(0) = \frac{\sum_{v'>0}(a_1(v',0) + a_2(v',0) \cdot O_2 + a_3(v',0) \cdot N_2 + (a_4(v',0) + a_5(v',0)) \cdot O) \cdot OH(v') + k_{18} \cdot O \cdot HO_2 + 2 \cdot k_{14} \cdot H \cdot HO_2}{k_{17} \cdot O},$$

$$HO_2 = \frac{k_{20} \cdot H \cdot M \cdot O_2}{k_{18} \cdot O + (k_{14} + k_{15} + k_{16}) \cdot H},$$

$$VER_{2\mu m} = a_1(9,7) \cdot OH(9) + a_1(8,9) \cdot OH(8),$$

$$VER_{1.6\mu m} = a_1(5,3) \cdot OH(5) + a_1(4,2) \cdot OH(4),$$

where $a_{1-6}$ are the constant rates of the processes OH($v$) → OH($v'<v$) + h$v$, OH($v$) +O$_2$ → OH($v<v'$) + O$_2$, OH($v$) +N$_2$ → OH($v<v'$) + N$_2$, OH($v$) +O($^3$P) → OH($v'≤v$-5) + O($^1$D), OH($v$) +O($^3$P)

$\rightarrow$ OH($v'<v$) + O($^3$P), and OH($v$) +O($^3$P) $\rightarrow$ H + O$_2$, respectively. Note this system includes 13 equations with 13 unknown variables. Therefore, the solution to the system for a set of the SABER data (simultaneously measured profiles of O$_3$, T, pressure, $VER_{2\mu m}$, and $VER_{1.6\mu m}$) gives one simultaneously retrieved profiles of O, H, OH($v$=0-9), and HO$_2$. By applying the criteria (17) and (25) to obtained O and H profiles, we verify the fulfillment of OH and HO$_2$ equilibrium conditions and determine the height below which the resulting profiles should be cut. More advanced retrieval procedure would be statistical, based on Bayesian theorem, taking into account the uncertainties in measurement data and rate constants. Similar, for example, to Kulikov et al. (2018a), it should include a derivation of posterior conditional probability density function of retrieved characteristics and numerical analysis of this function. Detailed development of this retrieval method is outside of this paper and should be carried out in a separate work.»

In addition, below we address a number of issues that attracted the Referee's criticism.

Specific Comments

\* The manuscript uses solar elevation angles greater than 95° to discriminate between twilight and nighttime conditions. This is insufficient and would include twilight conditions during which steady-state nighttime conditions may not have been established yet. The convention for astronomical nighttime is 108°, and various studies have used values in the range 100-110°. In comparison, the Panka et al. (2021) SABER data set this manuscript considers indicates a nighttime cutoff angle of 105° (this is according to the available data for 2009 at the website https://saber.gats-inc.com, the Panka et al. (2021) paper does not explicitly call this angle).

We apologize for our mistake and the misunderstanding it caused. Actually, all Figures in the paper were plotted using the lower threshold at 105° for the nighttime solar zenith angle (SZA). Nevertheless, during the revision stage, we plotted our results using different values of the SZA threshold and found that including twilight solar zenith angles in the range (95°-105°) allows extending the latitude range of OH and HO$_2$ equilibrium fulfillment (see Figures below and Supplement). When OH and HO$_2$ equilibrium conditions are applied in retrieval of unmeasured characteristics from measurement data, this also allows including noticeable part of data into the consideration.

[Figure]

*Nighttime mean and monthly averaged $HO_2/HO_2^{eq}$, where $HO_2^{eq}$ is equilibrium concentration determined by Eq. (4). Black line shows the border of $HO_2$ equilibrium according to condition (1). The stippling corresponds to $\chi<95°$. The white area represents the $<HO_2/HO_2^{eq}>$ ratio outside the [0.5, 1.5] interval.*

[Figure]

Nighttime mean and monthly averaged $OH/OH^{eq}$, where $OH^{eq}$ is equilibrium concentration determined by Eq. (5). Black line shows the border of OH equilibrium according to condition (1). The stippling corresponds to $\chi<95°$. The white area represents the $<OH/OH^{eq}>$ ratio outside the [0.5, 1.5] interval.

We are grateful to Referee for valuable remark and added a few sentences in this sense in the revised manuscript (see Discussion):

«Note the presented results were plotted using the lower threshold at 105° for the nighttime solar zenith angle ($\chi$) to exclude the twilight transition processes. Nevertheless, our additional analysis revealed that OH and $HO_2$ equilibrium conditions are well fulfilled at $\chi > 95°$. Evidently, during the processing of the measurement data, taking twilight $\chi$ in (95°,105°) range into account extends the latitude range of OH and $HO_2$ equilibrium application and includes noticeable part of data into the consideration. However, in this case, one should check for additional condition (Kulikov et al., 2023a):

$$e^{\int_{lt_{bn}}^{lt} \tau_{HO_2}^{-1} dt} \gg 1, \; e^{\int_{lt_{bn}}^{lt} \tau_{OH}^{-1} dt} \gg 1, \tag{27}$$

where $\tau_{HO_2}$ and $\tau_{OH}$ are the $HO_2$ and OH lifetimes determined by Eqs. (11) and (19), $lt$ is local time of data, $lt_{bn}$ is the local time at the beginning of the night. Note that at night, O and H tends to decrease due to the shutdown of the $O_x$ and $HO_x$ family photochemical sources. Thus, analyzing the measurement data, one can apply a more stringent conditions:

$$e^{\frac{lt-lt_{bn}}{\tau_{HO_2}}} \gg 1, \; e^{\frac{lt-lt_{bn}}{\tau_{OH}}} \gg 1. \tag{28}$$»

Also, note that 95° $\chi$ threshold was used in other papers. Please see, for example, Mlynczak et al. (JGR, 2013, https://doi.org/10.1002/jgrd.50401) and Mlynczak et al. (JGR, 2014, https://doi.org/10.1002/2013JD021263), where the nighttime ozone equilibrium condition was applied for O and H retrieval from SABER data. The Referee remark is more important for ozone, because its equilibrium concentration jumps at sunset due to the shutdown of $O_3$ photodissociation. Nevertheless, Kulikov et al. (ASR, 2023, https://doi.org/10.1016/j.asr.2022.11.005) confirmed using the SABER data that $\chi > 95°$ cutoff is enough to exclude the ozone in transition to a new equilibrium state. In the case of OH and $HO_2$, the photodissociation processes are not among main sinks and sources of these components in daytime (Kulikov et al., ACP, 2018, https://doi.org/10.5194/acp-18-7453-2018). Moreover, usually, the OH and $HO_2$ lifetimes, at least, during twilight time are essentially shorter than the $O_3$ lifetime (Kulikov et al., ASR, 2023, https://doi.org/10.1016/j.asr.2022.11.005).

* The criteria for equilibrium validity conditions require a difference of less than or equal to 10% between the species concentration calculated by the chemical transport model and that estimated using instantaneous local equilibrium between production and loss. The manuscript provides no discussion of uncertainties in the model, the observations, and several rate constants at the low MLT temperatures. To have confidence that when the equilibrium criterion is not met there is a meaningful deviation from equilibrium, all the other uncertainties must be significantly smaller than

the considered range of departure from equilibrium. This does not appear to be the case given all the above uncertainties. This is a critical point that is fundamental to this research effort. The comparison with the Panka et al.(2021) data set further demonstrates this point.

The Referee remark points to a very important issue, which was regrettably omitted in the initial version of the manuscript. That could, indeed, alter the perception of scientific significance of the results obtained.

The criterion for equilibrium condition validity of a certain trace gas limits a possible difference between local values of its true (in our case, calculated value) concentration $n$ and equilibrium concentration $n^{eq}$. Therefore, when equilibrium condition is applied to measurement data in the retrieval of unmeasured characteristics, the criterion allows to control and limit the possible error caused by the equilibrium approximation. Our analysis of data modeling for different trace gases shows that the ratio $n/n^{eq}$ in nonequilibrium areas varies widely and may rich several orders of magnitude.

[Figure]

*Examples of evolution of the ratio between local $O_3$ and OH concentrations and their equilibrium concentrations, respectively, obtained by the numerical simulation of the mesosphere – lower thermosphere.*

Thus, without using the proposed criteria or other restrictions (for example, in height or pressure), the error in retrieved characteristics due to the use of equilibrium approximation is uncontrollable and may significantly exceed all other errors in the retrieval procedure due to uncertainties of measurement data and rate constants. In the revised manuscript, we added a few sentences in this sense in Introduction:

«The local ratio between true and equilibrium concentrations may vary widely and reach up to several orders of magnitude (e.g., Figure 5 in Kulikov et al. (2018b)). Thus, without special restrictions, the error in retrieved characteristics due to the use of equilibrium approximation is

uncontrollable and may significantly exceed all other errors in the retrieval procedure due to, for example, uncertainties in the measurement data and rate constants.

……….

Note when equilibrium condition is applied to measurement data in the retrieval of unmeasured characteristics, the criterion allows controlling and limiting the possible error caused by the equilibrium approximation.»

Kulikov, M. Y., Belikovich, M. V., Grygalashvyly, M., Sonnemann, G. R., Ermakova, T. S., Nechaev, A. A., and Feigin, A. M.: Nighttime ozone chemical equilibrium in the mesopause region. J. Geophys. Res.,123, 3228–3242, https://doi.org/10.1002/2017JD026717, 2018b.

Moreover, following the Referee remark, we estimated the sensitivity of the presented $HO_2$ and OH criteria ($Crit_{HO_2}$ and $Crit_{OH}$) to the uncertainties of characteristics involving in the expressions of these criteria (in the revised manuscript, the conditions (17) and (25)). The local heights of the OH and $HO_2$ equilibrium borders ($z_{HO_2}{}^{crit}$ and $z_{OH}{}^{crit}$) according to the criteria are determined as the altitudes at which $Crit_{HO_2} = 0.1$ and $Crit_{OH} = 0.1$, respectively. We considered the whole dataset of nighttime profiles obtained by the numerical simulation of a one-year global evolution of mesosphere – lower thermosphere and evaluated total uncertainties to determination of $z_{HO_2}{}^{crit}$ and $z_{OH}{}^{crit}$ from each local (in time and space) dataset (profiles of O, H, $O_3$, M, $O_2$, and temperature). Following the typical analysis presented, for example, in Mlynczak et al. (JGR, 2013a, 2014), each uncertainty was calculated as a root sum square of the sensitivities to the individual perturbations of certain variables or parameters in the expressions (17) and (25). The following uncertainties for the variables were used: 5K in the temperature and 30% in $O_3$, O, and H. The uncertainties in rate constants with their temperature dependencies were taken from Burkholder et al. (2020). As the result, the monthly and longitudinally mean total uncertainties to determination of $z_{HO_2}{}^{crit}$ and $z_{OH}{}^{crit}$ were found varying in the range 0.02-1 km, depending on altitude and season.

[Figure]

*The monthly and longitudinally mean total uncertainties to determine the local heights of the OH and HO$_2$ equilibrium borders according to the criteria. The white area shows no data due to polar day.*

Note these values are comparable with the typical height resolution of satellite data that allows considering our criteria as a robust instrument for equilibrium condition validity. The main reason of relatively low sensitivity of $z_{HO_2}{}^{crit}$ and $z_{OH}{}^{crit}$ is the strong height-dependence of $Crit_{HO_2}$ and $Crit_{OH}$ near the value of 0.1.

[Figure]

*The examples of height-dependence of $Crit_{HO_2}$ and $Crit_{OH}$.*

In the revised manuscript, we added a Figure and a paragraph to address the issue (see Discussion):

«We evaluated the sensitivity of the presented HO$_2$ and OH criteria ($Crit_{HO_2}$ and $Crit_{OH}$) to the uncertainties of characteristics involving in the expressions (17) and (25). The local heights of the OH and HO$_2$ equilibrium borders ($z_{HO_2}{}^{crit}$ and $z_{OH}{}^{crit}$) according to the criteria are determined as the altitudes at which $Crit_{HO_2} = 0.1$ and $Crit_{OH} = 0.1$, respectively. We considered the whole dataset of nighttime profiles obtained by the numerical simulation of a one-year global evolution of

mesosphere – lower thermosphere and estimated total uncertainties to determination of $z_{HO_2}{}^{crit}$ and $z_{OH}{}^{crit}$ from each local (in time and space) dataset (profiles of O, H, $O_3$, M, $O_2$, and temperature). Following the typical analysis presented, for example, in Mlynczak et al. (2013a, 2014), each uncertainty was calculated as a root sum square of the sensitivities to the individual perturbations of certain variables or parameters in the expressions (17) and (25). The following uncertainties of the variables were used: 5K in the temperature and 30% in $O_3$, O, and H. The uncertainties in reaction rates and their temperature dependencies were taken from Burkholder et al. (2020). As the result (see Figure 7), the monthly and longitudinally mean total uncertainties to determination of $z_{HO_2}{}^{crit}$ and $z_{OH}{}^{crit}$ were found varying in the range 0.02-1 km, depending on altitude and season. Note these values are comparable with the typical height resolution of satellite data that allows considering our criteria as a robust instrument for equilibrium condition validity. The main reason of relatively low sensitivity of $z_{HO_2}{}^{crit}$ and $z_{OH}{}^{crit}$ is the strong height-dependence of $Crit_{HO_2}$ and $Crit_{OH}$ near the value of 0.1.»

* Regarding the examination of SABER remote-sensing observations and the modeling analysis by Panka et al.(2021), consideration of the relevant uncertainties would provide context and assess the results. There are substantial uncertainties in all the retrieved SABER minor constituents, [H], [O], and [O3] being the most relevant in this case. These are at best within approximately 20% and the relative errors are often largest at the lower and higher ends of the studied altitudes. With all this in mind, it seems that the data points shown in Figure 12 may well be outliers with limited significance at best. According to the manuscript (lines 406-407), "…the local height position of the OH equilibrium boundary can rise up to 87 km." The Panka et al. data set comprises 263,432 measurements and the number of points above 80 km is no more than approximately 2% (based on about five hundred points on each of the panels of Fig. 12). The overwhelming majority of those measurements are not consistent an equilibrium height above 80 km.

The number of points in the panels of the Figure 12 relative to the total number of profiles per a month varies in the range (3.3-7.7)%, depending on the month. Moreover, as was shown above, the OH and $HO_2$ equilibrium borders according to the criteria demonstrate low sensitivity to the uncertainties in the variables and parameters. Nevertheless, following this remark and the remark by the Referee 3, we have excluded the OH equilibrium boundary retrieval with the Panka et al. data from revised manuscript. As pointed by the Referee 3, the main reason is the question of the correctness of the Panka et al. approach for O retrieval below 85 km, see the discussion to the Kulikov et al. (ACP, 2023), pages 7-9 in https://egusphere.copernicus.org/preprints/2023/egusphere-2023-1046/egusphere-2023-1046-AC2-supplement.pdf. Additional question is the correctness of the Panka et al. approach for OH retrieval

below 87 km due to unknown source of $HO_2$ data. Instead of the deleted analysis, we give the detailed explanation in Discussion, how the results of our paper can help modifying the Panka et al. method for extending its capabilities:

«The results of our paper allow modifying the Panka et al. method to extend its capabilities. The simplest development of this method seems to be the following. First of all, note that the $HO_2$ equilibrium condition (9) depends on H and O only and can be used within the self-consistent retrieval procedure, considering the following system of equations:

$$OH(v = 1-9) = \frac{k_{12} \cdot H \cdot O_3 \cdot M \cdot f(v) + \sum_{v'>v}(a_1(v',v) + a_2(v',v) \cdot O_2 + a_3(v',v) \cdot N_2 + (a_4(v',v) + a_5(v',v)) \cdot O) \cdot OH(v')}{a_6(v)) \cdot O + \sum_{v>v'}(a_1(v,v') + a_2(v,v') \cdot O_2 + a_3(v,v') \cdot N_2 + (a_4(v,v') + a_5(v,v')) \cdot O)},$$

$$OH(0) = \frac{\sum_{v'>0}(a_1(v',0) + a_2(v',0) \cdot O_2 + a_3(v',0) \cdot N_2 + (a_4(v',0) + a_5(v',0)) \cdot O) \cdot OH(v') + k_{18} \cdot O \cdot HO_2 + 2 \cdot k_{14} \cdot H \cdot HO_2}{k_{17} \cdot O},$$

$$HO_2 = \frac{k_{20} \cdot H \cdot M \cdot O_2}{k_{18} \cdot O + (k_{14} + k_{15} + k_{16}) \cdot H},$$

$$VER_{2\mu m} = a_1(9,7) \cdot OH(9) + a_1(8,9) \cdot OH(8),$$

$$VER_{1.6\mu m} = a_1(5,3) \cdot OH(5) + a_1(4,2) \cdot OH(4),$$

where $a_{1-6}$ are the constant rates of the processes $OH(v) \rightarrow OH(v'<v) + hv$, $OH(v) + O_2 \rightarrow OH(v<v') + O_2$, $OH(v) + N_2 \rightarrow OH(v<v') + N_2$, $OH(v) + O(^3P) \rightarrow OH(v'\leq v-5) + O(^1D)$, $OH(v) + O(^3P) \rightarrow OH(v'<v) + O(^3P)$, and $OH(v) + O(^3P) \rightarrow H + O_2$, respectively. Note this system includes 13 equations with 13 unknown variables. Therefore, the solution to the system for a set of the SABER data (simultaneously measured profiles of $O_3$, T, pressure, $VER_{2\mu m}$, and $VER_{1.6\mu m}$) gives one simultaneously retrieved profiles of O, H, OH($v$=0-9), and $HO_2$. By applying the criteria (17) and (25) to obtained O and H profiles, we verify the fulfillment of OH and $HO_2$ equilibrium conditions and determine the height below which the resulting profiles should be cut. More advanced retrieval procedure would be statistical, based on Bayesian theorem, taking into account the uncertainties in measurement data and rate constants. Similar, for example, to Kulikov et al. (2018a), it should include a derivation of posterior conditional probability density function of retrieved characteristics and numerical analysis of this function. Detailed development of this retrieval method is outside of this paper and should be carried out in a separate work.»

\* This manuscript is for a technical note. The number of figures seems rather excessive (11 figures with 155 panels). The text does a good job describing the main point of the figures. It would seem practical to include most figures in a supplementary section.

Following this remark and the remark by the Referee 3, we organized the Supplement. In the revised manuscript, the Figures 3-8 were reduced to Figures 3-4. The complete figures for major reactions with all 12 panels for each month are presented in Supplement. Figure 12 was deleted. As

a result, the number of Figures was reduced from 12 to 8, and the total number of panels on them was reduced from 143 to 75.

\* Author Contributions: Co-authorship for the last three authors who "contributed to reviewing the article" seems somewhat unusual. Also, please note that the last initials should be AF and not AM (Line 434).

In the revised manuscript, Author Contributions was corrected as following:

«Conceptualization: MK, MB, AC, SD, AF. Methodology: MK, AF. Investigation: MK, MB, AC, SD. Software: MB, AC. Visualization: MB, AC. Funding Acquisition: MK. Writing – original draft preparation: MK, MB. Writing – review & editing: AC, SD. Supervising: AM.»

Technical Corrections

Line 19: …conditions describe the… …the top to some lower borders…

Line 21: …criteria reproduce well…

Line 22: …allow to extend previously proposed…

Line 35: …components: in particular, trace gases with short lifetimes relative to…

Line 47: …approximation has been used…

Line 79: …this approximation's validity

Line 80: …there is no assessment of…

Line 85: …maps of the components of interest.

Line 92: …apply this approach to the analysis…

Line 100: …all excited and ground states…

Line 128: …To remove the transition regions…

Line 146: …Finally, we obtained…

Line 147: …the poorly chemical evolution…???

Line 152: …from the total sink of n.

Line 169: …Figure 1 plots…

Line 173: …see the presence of…

Line 206: …Figure 3 shows…

Line 215: …Figure 4 presents…

Line 231: …up to a 95% contribution to the equilibrium concentration.

Line 243: …major in the lower part of…

Line 256: …are different from those…

Line 261: …As a result, the…

Lines 263: The punctuation (comma, in this case) appears to be incorporated in the equation. Similar comment for other equations. A few equations do not have punctuation. This issue could be resolved at the copy-editing stage.

Line 265: Figure 9 shows …

Line 271: …reproduces many features of…

Line 276: Figure 10 plots…

Line 288: Criteria for HO2 and…

Line 290: Let us determine…

Line 295: Let us find…

Line 297: "analyzed analytically" ???

Line 300: …rewritten in the following…

Line 316: "first of all" seems redundant

Line 317: …As previously mentioned, near and above the OH…

Line 331: There is a formatting problem with the equation.

Line 335: There is a formatting problem with the equation.

Line 339: …the red line...

Line 347: We will now discuss the obtained results…

Line 348: As noted, Figs. 9-10…

Line 349: Recently, Kulikov et al. (2023) found such a feature…

Line 355: "At middle," ???

Line 360: "From simplified Eqs. (17) and (25), it follows that…

Line 370: As a result,…

Line 372: There is a formatting problem with the equation.

Line 376: …including this additional term…

Line 378: As noted in the Introduction…

Line 379: …constitute a useful tool for retrieval of these components…

Line 384: …including improvement of existing retrieval…

Line 387: …valid for excited states…

Line 389: …lifetime is determined by the reaction…

Line 398: …the constant rates…

Line 406: …in accordance with… OR …according to…???

Line 439: Is the Acknowledgements section missing or is it the same as Financial Support?

Line 442: …and State assignment No. 0729…

Line 469: …a model study…

Line 511: SABER data…

Line 638: …averaged relative contribution…

Line 642: …averaged relative contribution…

Line 646: …averaged relative contribution…

Line 650: …averaged relative contribution…

Line 654: …averaged relative contribution…

Line 659: …averaged relative contribution…

All Referee's remarks were taken into account, the manuscript was corrected accordinary. Moreover, the final revised manuscript will be verified and corrected by a professional translator.

Other changes are related to the recommendations of other referees.

Thank you for taking your time to review our manuscript.

With respect,

Michael Kulikov, Michael Belikovich, Alexey Chubarov, Svetlana Dementyeva, and Alexander Feigin

---

## Author Response (AR1)

Dear Editor,

We would like to say many thanks the Referee for taking the time to review our manuscript and providing valuable recommendations. Their constructive criticism made the work clearer and more precise. We took into account all the remarks of Referee and, to the best of our ability, implemented the corresponding changes in the manuscript.

In the following, we address the comments point by point and show how the manuscript has been changed according to the comments. Below we use a certain color notation: comments by Referee are in red, our responses are in black, and the changes in the manuscript are in blue (placed inside the quotation marks).

**Response to the comments on the paper by Referee 1**

General Comments

.......

I am not clear on what significant new knowledge is gained. Some of the conclusions seem unsubstantiated and even unrealistic, especially the main conclusion that "the simultaneous application of OH and HO2 equilibrium to the SABER data (O3, volume emission rates at 2.0 and 1.6 um) together with the criteria (16) and (24) to control this equilibrium validity is going to retrieve all unknown HOx – Ox components (O, H, OH, and HO2), extending the altitude range of retrieval below 80 km and without external information."

In the revised manuscript, we give the detailed explanation of the noted conclusion (see lines 433-454 in Discussion):

«The results of our paper allow modifying the Panka et al. method to extend its capabilities. The simplest development of this method seems to be the following. First of all, note that the $HO_2$ equilibrium condition (9) depends on H and O only and can be used within the self-consistent retrieval procedure, considering the following system of equations:

$$OH(v=1-9) = \frac{k_{12} \cdot H \cdot O_3 \cdot M \cdot f(v) + \sum_{v'>v}(a_1(v',v) + a_2(v',v) \cdot O_2 + a_3(v',v) \cdot N_2 + (a_4(v',v) + a_5(v',v)) \cdot O) \cdot OH(v')}{a_6(v)) \cdot O + \sum_{v>v}'(a_1(v,v') + a_2(v,v') \cdot O_2 + a_3(v,v') \cdot N_2 + (a_4(v,v') + a_5(v,v')) \cdot O)},$$

$$OH(0) = \frac{\sum_{v'>0}(a_1(v',0) + a_2(v',0) \cdot O_2 + a_3(v',0) \cdot N_2 + (a_4(v',0) + a_5(v',0)) \cdot O) \cdot OH(v') + k_{18} \cdot O \cdot HO_2 + 2 \cdot k_{14} \cdot H \cdot HO_2}{k_{17} \cdot O},$$

$$HO_2 = \frac{k_{20} \cdot H \cdot M \cdot O_2}{k_{18} \cdot O + (k_{14} + k_{15} + k_{16}) \cdot H},$$

$$VER_{2\mu m} = a_1(9,7) \cdot OH(9) + a_1(8,9) \cdot OH(8),$$

$$VER_{1.6\mu m} = a_1(5,3) \cdot OH(5) + a_1(4,2) \cdot OH(4),$$

where $a_{1-6}$ are the constant rates of the processes $OH(v) \rightarrow OH(v'<v) + hv$, $OH(v) + O_2 \rightarrow OH(v<v') + O_2$, $OH(v) + N_2 \rightarrow OH(v<v') + N_2$, $OH(v) + O(^3P) \rightarrow OH(v' \leq v-5) + O(^1D)$, $OH(v) + O(^3P)$

$\rightarrow$ OH($v'<v$) + O($^3$P), and OH($v$) +O($^3$P) $\rightarrow$ H + O$_2$ respectively. Take into consideration, that this system includes 13 equations with 13 unknown variables. Therefore, the solution to the system for a single set of the SABER measurements (simultaneously measured profiles of O$_3$, T, pressure, $VER_{2\mu m}$, and $VER_{1.6\mu m}$) gives one simultaneously retrieved profiles of O, H, OH($v$=0-9), and HO$_2$. By applying the criteria (17) and (25) to obtained O and H profiles, we verify the fulfillment of OH and HO$_2$ equilibrium conditions and determine the height, below which the resulting profiles should be cut. More advanced retrieval procedure would be statistical, based on Bayesian theorem, taking into account the uncertainties in measurement data and rate constants. Similarly, for example, to Kulikov et al. (2018a), it should include a derivation of posterior conditional probability density function of retrieved characteristics and numerical analysis of this function. Detailed development of this retrieval method is outside of this paper and should be carried out in a separate work.»

In addition, below we address a number of issues that attracted the Referee's criticism.

Specific Comments

\* The manuscript uses solar elevation angles greater than 95° to discriminate between twilight and nighttime conditions. This is insufficient and would include twilight conditions during which steady-state nighttime conditions may not have been established yet. The convention for astronomical nighttime is 108°, and various studies have used values in the range 100-110°. In comparison, the Panka et al. (2021) SABER data set this manuscript considers indicates a nighttime cutoff angle of 105° (this is according to the available data for 2009 at the website https://saber.gats-inc.com, the Panka et al. (2021) paper does not explicitly call this angle).

We apologize for our mistake and the misunderstanding it caused. Actually, all Figures in the paper were plotted using the lower threshold at 105° for the nighttime solar zenith angle (SZA). Nevertheless, during the revision stage, we plotted our results using different values of the SZA threshold and found that including twilight solar zenith angles in the range (95°-105°) allows extending the latitude range of OH and HO$_2$ equilibrium fulfillment (see Figures below and Supplement). When OH and HO$_2$ equilibrium conditions are applied in retrieval of unmeasured characteristics from measurement data, this also allows including noticeable part of data into the consideration.

[Figure]

*Nighttime mean and monthly averaged $HO_2/HO_2{}^{eq}$, where $HO_2{}^{eq}$ is equilibrium concentration determined by Eq. (4). Black line shows the border of $HO_2$ equilibrium according to condition (1). The stippling corresponds to $\chi < 95°$. The white area represents the $< HO_2/HO_2{}^{eq} >$ ratio outside the [0.5, 1.5] interval.*

[Figure]

Nighttime mean and monthly averaged $OH/OH^{eq}$, where $OH^{eq}$ is equilibrium concentration determined by Eq. (5). Black line shows the border of OH equilibrium according to condition (1). The stippling corresponds to $\chi<95°$. The white area represents the $<OH/OH^{eq}>$ ratio outside the [0.5, 1.5] interval.

We are grateful to Referee for valuable remark and added a few sentences in this sense in the revised manuscript (see lines 352-364 in Discussion):

«Pay attention to the fact, that the presented results were plotted, using the lower threshold at 105° for the nighttime solar zenith angle ($\chi$) to exclude the twilight transition processes. Nevertheless, our additional analysis revealed, that OH and $HO_2$ equilibrium conditions are fulfilled at $\chi>95°$. Evidently, during the processing of the measurement data, taking twilight $\chi$ in (95°,105°) range into account extends the latitude range of OH and $HO_2$ equilibria application and allows us to include a noticeable part of the data into consideration. However, in this case one should check for additional condition (Kulikov et al., 2023a):

$$e^{\int_{lt_{bn}}^{lt} \tau_{HO_2}^{-1} dt} \gg 1, \, e^{\int_{lt_{bn}}^{lt} \tau_{OH}^{-1} dt} \gg 1, \tag{27}$$

where $\tau_{HO_2}$ and $\tau_{OH}$ are the $HO_2$ and OH lifetimes, determined by Eqs. (11) and (19), $lt$ is local time of data, $lt_{bn}$ is the local time at the beginning of the night. Mind, that at night O and H tend to decrease due to the shutdown of the $O_x$ and $HO_x$ family photochemical sources, so $\tau_{HO_2}$ and $\tau_{OH}$ increase. Thus, analyzing the measurement data one can apply more stringent conditions:

$$e^{\frac{lt-lt_{bn}}{\tau_{HO_2}}} \gg 1, \, e^{\frac{lt-lt_{bn}}{\tau_{OH}}} \gg 1. \tag{28}$$»

Also, note that 95° $\chi$ threshold was used in other papers. Please see, for example, Mlynczak et al. (JGR, 2013, https://doi.org/10.1002/jgrd.50401) and Mlynczak et al. (JGR, 2014, https://doi.org/10.1002/2013JD021263), where the nighttime ozone equilibrium condition was applied for O and H retrieval from SABER data. The Referee remark is more important for ozone, because its equilibrium concentration jumps at sunset due to the shutdown of $O_3$ photodissociation. Nevertheless, Kulikov et al. (ASR, 2023, https://doi.org/10.1016/j.asr.2022.11.005) confirmed using the SABER data that $\chi>95°$ cutoff is enough to exclude the ozone in transition to a new equilibrium state. In the case of OH and $HO_2$, the photodissociation processes are not among main sinks and sources of these components in daytime (Kulikov et al., ACP, 2018, https://doi.org/10.5194/acp-18-7453-2018). Moreover, usually, the OH and $HO_2$ lifetimes, at least, during twilight time are essentially shorter than the $O_3$ lifetime (Kulikov et al., ASR, 2023, https://doi.org/10.1016/j.asr.2022.11.005).

* The criteria for equilibrium validity conditions require a difference of less than or equal to 10% between the species concentration calculated by the chemical transport model and that estimated using instantaneous local equilibrium between production and loss. The manuscript provides no discussion of uncertainties in the model, the observations, and several rate constants at the low MLT temperatures. To have confidence that when the equilibrium criterion is not met there is a meaningful deviation from equilibrium, all the other uncertainties must be significantly smaller than

the considered range of departure from equilibrium. This does not appear to be the case given all the above uncertainties. This is a critical point that is fundamental to this research effort. The comparison with the Panka et al.(2021) data set further demonstrates this point.

The Referee remark points to a very important issue, which was regrettably omitted in the initial version of the manuscript. That could, indeed, alter the perception of scientific significance of the results obtained.

The criterion for equilibrium condition validity of a certain trace gas limits a possible difference between local values of its true (in our case, calculated value) concentration $n$ and equilibrium concentration $n^{eq}$. Therefore, when equilibrium condition is applied to measurement data in the retrieval of unmeasured characteristics, the criterion allows to control and limit the possible error caused by the equilibrium approximation. Our analysis of data modeling for different trace gases shows that the ratio $n/n^{eq}$ in nonequilibrium areas varies widely and may rich several orders of magnitude.

[Figure]

*Examples of evolution of the ratio between local $O_3$ and OH concentrations and their equilibrium concentrations, respectively, obtained by the numerical simulation of the mesosphere – lower thermosphere.*

Thus, without using the proposed criteria or other restrictions (for example, in height or pressure), the error in retrieved characteristics due to the use of equilibrium approximation is uncontrollable and may significantly exceed all other errors in the retrieval procedure due to uncertainties of measurement data and rate constants. In the revised manuscript, we added a few sentences in this sense in Introduction, see lines 82-86 and 96-98:

[revised manuscript text omitted]

\* Regarding the examination of SABER remote-sensing observations and the modeling analysis by Panka et al.(2021), consideration of the relevant uncertainties would provide context and assess the results. There are substantial uncertainties in all the retrieved SABER minor constituents, [H], [O], and [O3] being the most relevant in this case. These are at best within approximately 20% and the relative errors are often largest at the lower and higher ends of the studied altitudes. With all this in mind, it seems that the data points shown in Figure 12 may well be outliers with limited significance at best. According to the manuscript (lines 406-407), "…the local height position of the OH equilibrium boundary can rise up to 87 km." The Panka et al. data set comprises 263,432 measurements and the number of points above 80 km is no more than approximately 2% (based on about five hundred points on each of the panels of Fig. 12). The overwhelming majority of those measurements are not consistent an equilibrium height above 80 km.

The number of points in the panels of the Figure 12 relative to the total number of profiles per a month varies in the range (3.3-7.7)%, depending on the month. Moreover, as was shown above, the OH and HO$_2$ equilibrium borders according to the criteria demonstrate low sensitivity to the uncertainties in the variables and parameters. Nevertheless, following this remark and the remark by the Referee 3, we have excluded the OH equilibrium boundary retrieval with the Panka et al. data from revised manuscript. As pointed by the Referee 3, the main reason is the question of the correctness of the Panka et al. approach for O retrieval below 85 km, see the discussion to the Kulikov et al. (ACP, 2023), pages 7-9 in https://egusphere.copernicus.org/preprints/2023/egusphere-2023-1046/egusphere-2023-1046-AC2-

supplement.pdf. Additional question is the correctness of the Panka et al. approach for OH retrieval below 87 km due to unknown source of $HO_2$ data. Instead of the deleted analysis, we give the detailed explanation in Discussion (see lines 433-454), how the results of our paper can help modifying the Panka et al. method for extending its capabilities:

«The results of our paper allow modifying the Panka et al. method to extend its capabilities. The simplest development of this method seems to be the following. First of all, note that the $HO_2$ equilibrium condition (9) depends on H and O only and can be used within the self-consistent retrieval procedure, considering the following system of equations:

$$OH(v = 1 - 9) = \frac{k_{12} \cdot H \cdot O_3 \cdot M \cdot f(v) + \sum_{v'>v}(a_1(v',v) + a_2(v',v) \cdot O_2 + a_3(v',v) \cdot N_2 + (a_4(v',v) + a_5(v',v)) \cdot O) \cdot OH(v')}{a_6(v)) \cdot O + \sum_{v>v}'(a_1(v,v') + a_2(v,v') \cdot O_2 + a_3(v,v') \cdot N_2 + (a_4(v,v') + a_5(v,v')) \cdot O)},$$

$$OH(0) = \frac{\sum_{v'>0}(a_1(v',0) + a_2(v',0) \cdot O_2 + a_3(v',0) \cdot N_2 + (a_4(v',0) + a_5(v',0)) \cdot O) \cdot OH(v') + k_{18} \cdot O \cdot HO_2 + 2 \cdot k_{14} \cdot H \cdot HO_2}{k_{17} \cdot O},$$

$$HO_2 = \frac{k_{20} \cdot H \cdot M \cdot O_2}{k_{18} \cdot O + (k_{14} + k_{15} + k_{16}) \cdot H},$$

$$VER_{2\mu m} = a_1(9,7) \cdot OH(9) + a_1(8,9) \cdot OH(8),$$

$$VER_{1.6\mu m} = a_1(5,3) \cdot OH(5) + a_1(4,2) \cdot OH(4),$$

where $a_{1-6}$ are the constant rates of the processes $OH(v) \rightarrow OH(v'{<}v) + hv$, $OH(v) + O_2 \rightarrow OH(v{<}v') + O_2$, $OH(v) + N_2 \rightarrow OH(v{<}v') + N_2$, $OH(v) + O(^3P) \rightarrow OH(v'{\leq}v{-}5) + O(^1D)$, $OH(v) + O(^3P) \rightarrow OH(v'{<}v) + O(^3P)$, and $OH(v) + O(^3P) \rightarrow H + O_2$ respectively. Take into consideration, that this system includes 13 equations with 13 unknown variables. Therefore, the solution to the system for a single set of the SABER measurements (simultaneously measured profiles of $O_3$, T, pressure, $VER_{2\mu m}$, and $VER_{1.6\mu m}$) gives one simultaneously retrieved profiles of O, H, OH($v$=0-9), and $HO_2$. By applying the criteria (17) and (25) to obtained O and H profiles, we verify the fulfillment of OH and $HO_2$ equilibrium conditions and determine the height, below which the resulting profiles should be cut. More advanced retrieval procedure would be statistical, based on Bayesian theorem, taking into account the uncertainties in measurement data and rate constants. Similarly, for example, to Kulikov et al. (2018a), it should include a derivation of posterior conditional probability density function of retrieved characteristics and numerical analysis of this function. Detailed development of this retrieval method is outside of this paper and should be carried out in a separate work.»

\* This manuscript is for a technical note. The number of figures seems rather excessive (11 figures with 155 panels). The text does a good job describing the main point of the figures. It would seem practical to include most figures in a supplementary section.

Following this remark and the remark by the Referee 3, we organized the Supplement. In the revised manuscript, the Figures 3-8 were reduced to Figures 3-4. The complete figures for major reactions with all 12 panels for each month are presented in Supplement. Figure 12 was deleted. As

a result, the number of Figures was reduced from 12 to 8, and the total number of panels on them was reduced from 143 to 75.

* Author Contributions: Co-authorship for the last three authors who "contributed to reviewing the article" seems somewhat unusual. Also, please note that the last initials should be AF and not AM (Line 434).

In the revised manuscript, Author Contributions was corrected as following:

«Conceptualization: MK, MB, AC, SD, AF. Methodology: MK, AF. Investigation: MK, MB, AC, SD. Software: MB, AC. Visualization: MB, AC. Funding Acquisition: MK. Writing – original draft preparation: MK, MB. Writing – review & editing: AC, SD. Supervising: AF.»

Technical Corrections

Line 19: …conditions describe the… …the top to some lower borders…

Line 21: …criteria reproduce well…

Line 22: …allow to extend previously proposed…

Line 35: …components: in particular, trace gases with short lifetimes relative to…

Line 47: …approximation has been used…

Line 79: …this approximation's validity

Line 80: …there is no assessment of…

Line 85: …maps of the components of interest.

Line 92: …apply this approach to the analysis…

Line 100: …all excited and ground states…

Line 128: …To remove the transition regions…

Line 146: …Finally, we obtained…

Line 147: …the poorly chemical evolution…???

Line 152: …from the total sink of n.

Line 169: …Figure 1 plots…

Line 173: …see the presence of…

Line 206: …Figure 3 shows…

Line 215: …Figure 4 presents…

Line 231: …up to a 95% contribution to the equilibrium concentration.

Line 243: …major in the lower part of…

Line 256: …are different from those…

Line 261: …As a result, the…

Lines 263: The punctuation (comma, in this case) appears to be incorporated in the equation. Similar comment for other equations. A few equations do not have punctuation. This issue could be resolved at the copy-editing stage.

Line 265: Figure 9 shows …

Line 271: …reproduces many features of…

Line 276: Figure 10 plots…

Line 288: Criteria for HO2 and…

Line 290: Let us determine…

Line 295: Let us find…

Line 297: "analyzed analytically" ???

Line 300: …rewritten in the following…

Line 316: "first of all" seems redundant

Line 317: …As previously mentioned, near and above the OH…

Line 331: There is a formatting problem with the equation.

Line 335: There is a formatting problem with the equation.

Line 339: …the red line...

Line 347: We will now discuss the obtained results…

Line 348: As noted, Figs. 9-10…

Line 349: Recently, Kulikov et al. (2023) found such a feature…

Line 355: "At middle," ???

Line 360: "From simplified Eqs. (17) and (25), it follows that…

Line 370: As a result,…

Line 372: There is a formatting problem with the equation.

Line 376: …including this additional term…

Line 378: As noted in the Introduction…

Line 379: …constitute a useful tool for retrieval of these components…

Line 384: …including improvement of existing retrieval…

Line 387: …valid for excited states…

Line 389: …lifetime is determined by the reaction…

Line 398: …the constant rates…

Line 406: …in accordance with… OR …according to…???

Line 439: Is the Acknowledgements section missing or is it the same as Financial Support?

Line 442: …and State assignment No. 0729…

Line 469: …a model study…

Line 511: SABER data…

Line 638: …averaged relative contribution…

Line 642: …averaged relative contribution…

Line 646: …averaged relative contribution…

Line 650: …averaged relative contribution…

Line 654: …averaged relative contribution…

Line 659: …averaged relative contribution…

All Referee's remarks were taken into account, the manuscript was corrected accordinary. Moreover, the revised manuscript has been verified and corrected by a professional translator (American style).

Other changes are related to the recommendations of other referees.

Thank you for taking your time to review our manuscript.

With respect,

Michael Kulikov, Michael Belikovich, Alexey Chubarov, Svetlana Dementyeva, and Alexander Feigin

Dear Editor,

We would like to say many thanks the Referee for taking the time to review our manuscript and providing valuable recommendations. Their constructive criticism made the work clearer and more precise. We took into account all the remarks of Referee and, to the best of our ability, implemented the corresponding changes in the manuscript.

In the following, we address the comments point by point and show how the manuscript has been changed according to the comments. Below we use a certain color notation: comments by Referee are in red, our responses are in black, and the changes in the manuscript are in blue (placed inside the quotation marks).

**Response to the comments on the paper by Referee 2**

Major comment

The approach describes, evaluates, and compares several expressions to determine equilibrium. The standard for evaluation is how well a particular equilibrium value computed from their 3-D chemical transport model agrees with the actual concentrations simulated in the same model. Winds used for transport and temperatures for chemical rate coefficients are based on temporal smoothing of once-daily values from a middle atmosphere dynamics model. My concern is that this approach excludes the transport and large temperature swings associated with tides. In the tropical MLT, vertical winds associated with the migrating diurnal tide can be quite substantial and are a leading transport process. Temperature variations of 10-20 K in a few hours are seen during some seasons. As a result of this omission, the actual variations of species concentrations that go into the analyses may be more variable than those simulated, which would affect the standard deviation criteria in Eq. (1). This omission may have led to a diagnosis of equilibrium that is more optimistic than the reality for the equatorial region.

It seems to me that the only way to quantify the importance of tides would be to perform a simulation with input dynamical fields taken more frequently. Without this, it is necessary to add some sentences that point out this omission and its possible implications for the results.

During the revision period, we carried out additional modelling with 3D distributions of the main characteristics taken from the Canadian Middle Atmosphere Model (CMAM) for the year 2009 (Scinocca et al., 2008) updated with the 6 hour of frequency. CMAM is known to reproduce tides e.g. McLandress, 1997; Jonsson et al., 2002). The analysis of the height-time evolution of OH and HO$_2$, especially in the tropical latitudes, showed that our criteria well reproduce the changes of the OH and HO$_2$ equilibrium boundaries in such conditions (see Figure below).

Scinocca, J. F., McFarlane, N. A., Lazare, M., Li, J., and Plummer, D.: The CCCma third generation AGCM and its extension into the middle atmosphere, Atmos. Chem. Phys., 8, 7055–7074, https://doi.org/10.5194/acp-8-7055-2008, 2008.

McLandress, C.: Seasonal variability of the diurnal tide: Results from the Canadian Middle Atmosphere General Circulation Model, J. Geophys. Res., 102, 29 747–29 764, 1997.

Jonsson, A., de Grandpr´e, J. and McConnell, J. C.: A comparison of mesospheric temperatures from the Canadian Middle Atmosphere Model and HALOE observations: zonal mean and signature

of the solar diurnal tide, Geophys. Res. Lett., 29(9), 1346, doi:10.1029/2001GL014476, 2002.

[Figure]

$HO_2/HO_{2sh}^{eq}$ *(two top panels) and* $OH/OH_{sh}^{eq}$ *(two bottom panels) time-height variations above the Equator (2.8°S,0°W) in March and June 2009 calculated with the use of the temperature and winds distributions from the Canadian Middle Atmosphere Model. The stippling shows daytime. The white area represents the* $HO_2/HO_{2sh}^{eq}$ *and* $OH/OH_{sh}^{eq}$ *ratios outside the [0.5, 1.5] interval. Magenta lines point the borders of HO₂ and OH equilibrium according to criteria (17) and (25) ($Crit_{HO_2} = 0.1$ and $Crit_{OH} = 0.1$).*

This Figure can be found in the Supplement (Figure S27). In the Discussion of the revised manuscript, we have added the following paragraph to address the issue (see lines 365-372):

«The main results were obtained using a 3D model, where temperature and wind distributions are updated every 24 hours. This excluded the influence of the atmospheric wave motion, in particular, associated with tides, which is one of the main dynamical drivers in the tropical mesopause. We carried out additional modeling with the distributions of the main characteristics, calculated by the Canadian Middle Atmosphere Model for the year 2009 (Scinocca et al., 2008) with a 6-hourly frequency for updating. The analysis of the time-height evolution of OH and $HO_2$, especially at low latitudes, showed that our criteria reproduce quite well the local variations of the OH and $HO_2$ equilibrium boundaries in such conditions.»

Other comments

(Figure captions) In some cases, the captions and text do not explain the figures sufficiently. In particular, the captions to Figures 1 and 9 are almost identical; the exception being a subscript "sh" in one of the terms in the Figure 9 caption. The situation for Figures 2 and 10 is similar. Please add words giving more information so the differences are more obvious.

In the revised manuscript, these Figure captions were corrected accordingly:

«Figure 1. Nighttime mean and monthly averaged $HO_2/HO_2^{eq}$, where $HO_2^{eq}$ is equilibrium concentration determined by Eq. (4). Black line shows the border of $HO_2$ equilibrium according to condition (1). The stippling corresponds to $\chi<105°$. The white area represents the $< HO_2/HO_2^{eq} >$ ratio outside the [0.5, 1.5] interval.

Figure 2. Nighttime mean and monthly averaged $OH/OH^{eq}$, where $OH^{eq}$ is equilibrium concentration determined by Eq. (5). Black line shows the border of OH equilibrium according to condition (1). The stippling corresponds to $\chi<105°$. The white area represents the $< OH/OH^{eq} >$ ratio outside the [0.5, 1.5] interval.

Figure 3. Nighttime mean and monthly averaged relative contribution of a certain reaction to the total source or sink of $HO_2$ in equilibrium areas. The stippling corresponds to $\chi<105°$. White color indicates nonequilibrium areas of $HO_2$.

Figure 4. Nighttime mean and monthly averaged relative contribution of a certain reaction to the total source or sink of OH in equilibrium areas. The stippling corresponds to $\chi<105°$. White color indicates nonequilibrium areas of OH.

Figure 5. Nighttime mean and monthly averaged $HO_2/HO_{2sh}^{eq}$, where $HO_{2sh}^{eq}$ is shortened equilibrium concentration determined by Eq. (9). Black line shows the border of $HO_2$ equilibrium according to condition (1). Magenta line shows $< Crit_{HO_2} >= 0.1$. The stippling corresponds to $\chi<105°$. The white area represents the $< HO_2/HO_2^{eq} >$ ratio outside the [0.5, 1.5] interval.

Figure 6. Nighttime mean and monthly averaged $OH/OH_{sh}^{eq}$, where $OH_{sh}^{eq}$ is shortened equilibrium concentration determined by Eq. (10). Black line shows the border of OH equilibrium according to condition (1). Magenta line shows $< Crit_{OH} >= 0.1$. The stippling corresponds to $\chi<105°$. The white area represents the $< OH/OH^{eq} >$ ratio outside the [0.5, 1.5] interval.»

(line 43-46) Can you provide references or more detail to support the idea that limited measurements of trace species can be used to retrieve temperature, reaction rates, chemical sources, etc? This comes across as wishful thinking that might not hold up because of multiple uncertainties in the components of the photochemical system.

In the revised manuscript, this sentence has been corrected as follows (see lines 45-49):

«These relationships can be used to derive information about hard-to-measure atmospheric species, determine key atmospheric characteristics (for example, temperature (Marchand et al., 2007)), validate the data quality of simultaneous measurements of several atmospheric components (Kulikov et al., 2018a), estimate reaction rate constants (Stedman et al., 1975; Avallone and Toohey, 2001), evaluate sources/sinks (Cantrell et al., 2003), etc.»

Stedman, D. H., Chameides, W., and Jackson, J. O.: Comparison of experimental and computed values for J(NO2), Geophys. Res. Lett., 2, 22–25, https://doi.org/10.1029/GL002i001p00022, 1975.

Avallone, L. M. and Toohey, D. W.: Tests of halogen photochemistry using in situ measurements of ClO and BrO in the lower polar stratosphere, J. Geophys. Res., 106, 10411–1042, https://doi.org/10.1029/2000JD900831, 2001.

Cantrell, C. A., et al.: Steady state free radical budgets and ozone photochemistry during TOPSE, J. Geophys. Res., 108, TOP9-1–TOP9-22, https://doi.org/10.1029/2002JD002198, 2003.

Marchand, M., Bekki, S., Lefevre, F., and Hauchecorne, A.: Temperature retrieval from stratospheric O3 and NO3 GOMOS data, Geophys. Res. Lett., 34, L24809, https://doi.org/10.1029/2007GL030280, 2007.

Kulikov, M. Y., Nechaev, A. A., Belikovich, M. V., Ermakova, T. S., and Feigin, A. M.: Technical note: Evaluation of the simultaneous measurements of mesospheric OH, $HO_2$, and $O_3$ under a photochemical equilibrium assumption – a statistical approach, Atm. Chem. Phys., 18, 7453-747, https://doi.org/10.5194/acp-18-7453-2018, 2018a.

(line 78-81) The two sentences (beginning "Secondly, there is no detailed numerical evaluation ..") are confusing. This paragraph is about night ozone, which you examined in earlier papers. Have you switched the discussion to HOx without informing the reader or are you raising doubts about your 2019 and 2023 papers on ozone equilibrium?

In the revised manuscript, these sentences have been replaced as follows (see lines 82-86):

«The local ratio between true and equilibrium concentrations may vary widely and reach up to several orders of magnitude (e.g., Figure 5 in Kulikov et al. (2018b)). Thus, without special restrictions the error in retrieved characteristics due to the use of equilibrium approximation is uncontrollable and may significantly exceed all other errors in the retrieval procedure due to, for example, uncertainties in the measurement data and rate constants.»

Kulikov, M. Y., Belikovich, M. V., Grygalashvyly, M., Sonnemann, G. R., Ermakova, T. S., Nechaev, A. A., and Feigin, A. M.: Nighttime ozone chemical equilibrium in the mesopause region. J. Geophys. Res.,123, 3228–3242, https://doi.org/10.1002/2017JD026717, 2018b.

(line 101) "this approach is tested" It seems that you test the equilibrium timescale for ground-state OH but not for the vibrationally excited states that are important components in the model of Panka et al. Please revise to make this distinction clear.

In the revised manuscript, this paragraph has been corrected as follows (see lines 105-108):

«In particular, Panka et al. (2021) proposed the method for nighttime total OH retrieval from SABER/TIMED data at 80-100 km, which does not use the ozone chemical equilibrium. However, the method applies the equilibrium between sources and sinks not only to excited states of OH with ultrashort lifetimes, but also to the ground state. Therefore, this point is verified in our paper.»

(Section 6) It was difficult to get oriented toward this analysis. Please add a sentence or more at the beginning of the section referring the reader to Eq (2) and also please reiterate the key takeaway from the discussion describing the difference between the terms "lifetime" and "local time scale". Other readers may, like me, be unfamiliar with the distinction between these concepts and their role in your analysis.

In the revised manuscript, the Section 2 «Used 3D model and Approaches» has been extended to address this remark (see lines 152-169):

«Finally, we obtained and verified the analytical criteria of OH and $HO_2$ nighttime chemical equilibria according to Kulikov et al. (2023a). The paper considered the pure chemical evolution of a certain trace gas $n$:

$$\frac{dn}{dt} = I_n - S_n = -\frac{1}{\tau_n}(n - n^{eq}),$$

$$\tau_n = \frac{n}{S_n}, \ n^{eq} = \frac{n \cdot I_n}{S_n}, \tag{2}$$

where $t$ is time, $I_n$ and $S_n$ are total photochemical/chemical sources and sinks of $n$ respectively, $\tau_n$ is the $n$ lifetime and $n^{eq}$ is its equilibrium concentration, corresponding to the condition $I_n = S_n$. The lifetime determines the characteristic time scale, for which $n$ approaches $n^{eq}$, when $n^{eq} =$

$const.$ In general case $\tau_n$ and $n^{eq}$ are functions of time. Kulikov et al. (2023a) showed strictly mathematically, that the local values of $n$ and $n^{eq}$ are close to each other ($n(t) \approx n^{eq}(t)$), when $\tau_n \ll \tau_{n^{eq}}$, where $\tau_{n^{eq}}$ is the local time scale of $n^{eq}$:

$$\tau_{n^{eq}} \equiv \frac{n^{eq}}{|dn^{eq}/dt|}. \tag{3}$$

The expression for $\tau_n$ is found from the total sink of $n$. The expression for $\tau_{n^{eq}}$ is derived from Eq. (3) with the use of differential equations, describing chemical evolution of other reacting components, which determine the expression for $n^{eq}$. Kulikov et al. (2023a) also showed, when $\tau_n \ll \tau_{n^{eq}}$, $n \cong n^{eq}(1 - sign(\frac{dn^{eq}}{dt}) \cdot \frac{\tau_n}{\tau_{n^{eq}}})$ in the first order approximation. Thus, the criterion

$$\tau_n/\tau_{n^{eq}} \leq 0.1 \tag{4}$$

is sufficient, in order to the possible relative difference between $n$ and $n^{eq}$ to be no more than 0.1.»

Is the analysis of ozone equilibrium in Section 7 (lines 348-362) relevant to the present paper? If so, please explain the connection. If this paragraph remains, please provide some transition words to let the reader know when you are switching your focus from ozone to HOx.

In the revised manuscript, this paragraph has been corrected as follows (see lines 390-401):

«As noted, Figs. 5-6 represent an interesting peculiarity. At the middle latitudes summer $z_{HO_{2sh}^{eq}}$ and $z_{OH_{sh}^{eq}}$ are remarkably higher than winter ones. Recently, Kulikov et al. (2023b) found such a feature in the evolution of nighttime ozone chemical equilibrium boundary, derived from SABER/TIMED data, which was accompanied by the same variation of the transition zone, separating deep and weak photochemical oscillations of O and H, caused by the diurnal variations of solar radiation. The authors analyzed this effect near and below the transition zone. It was shown firstly, that nighttime O decreases with the characteristic time scale $\tau_O = O/|dO/dt|$ proportional to the $O/H$ ratio at the beginning of the night. Secondly, during the summer the daytime $O/H$ at the middle latitudes is remarkably less than the one in winter. Consequently, summer values of $\tau_O$ are significantly shorter than winter ones, so summer O during the night decreases much faster than in winter. In our case lifetimes of $HO_2$ and OH are proportional mainly to $\frac{1}{O}$ (see Eqs. (11) and (19)), so the summer rise of $z_{HO_{2sh}^{eq}}$ and $z_{OH_{sh}^{eq}}$ can be explained by the season difference in O diurnal evolution at these latitudes. »

(line 391) "one can see from Figure 8" Should this be Figure 7 or Figures 7 and 8?

In the revised manuscript, it should be Figure 6. Corrected.

Editorial comments

Throughout: I'm not sure of the journal's style guidelines but, for me as a reader, it would be really helpful if you indicated that something is an equation when the number appears in the text. For example (line 141), replace "from 1" with "from Eq. (1)" or something similar.

(line 51) "of a critical parameters" -> "of critical parameters"

(line 66) Sentence beginning "First …" is not clear. Is this what you mean? "First, there are no clear criteria indicating the conditions under which the equilibrium conditions are satisfied?"

(line 71) "is too short varying" -> "varies"

(line 83) "to correct search of" -> "to correctly search for"

(line 100) "exited" -> "excited"

(line 128) "we took into account the local time" -> "we use only local times"

(line 147) "the poorly chemical evolution" It is not clear what this means.

(line 170) "dashed area" -> "stippling" Also this sentence would fit better in the figure caption than the main text.

(line 173) "present" -> "presence"

(line 263) apostrophe in denominator

(line 288) "criterions" -> "criteria"

(line 290, 295) "Let" is an awkward word here. How about replace the first instance (line 290) with "First, we" and the second (line 295) with "Then"

(line 310, 343) "in zero approximation" do you mean "in the zeroth order approximation"?

(line 351) "whichwas" -> "which was"

(line 352-355) This sentence is too long and convoluted and the point being made is not clear. What are "nighttime evolution times"? In the next sentence, what does "At middle" refer to?

(line 360) "It is follows" -> "It follows"

(line 426) "is going to retrieve" Do you mean "allows the retrieval of"?

All Referee remarks have been taken into account and the manuscript has been corrected accordingly. Moreover, the revised manuscript has been verified and corrected by a professional translator (American style).

Other changes are related to the recommendations of other referees.

Thank you for taking your time to review our manuscript.

With respect,

Michael Kulikov, Michael Belikovich, Alexey Chubarov, Svetlana Dementyeva, and Alexander Feigin

Dear Editor,

We would like to say many thanks the Referee for taking the time to review our manuscript and valuable recommendations. We have tried to follow the referee remarks and have utilized all of them.

In the following, we address the comments point by point and show how the manuscript has been changed accordingly to the comments. Below comments by Referees are in red, our responses are in black, the changes in the manuscript are in blue and brackets.

**Response to the comments on the paper by Referee 3**

1. By the meaning, construction, and titles the work is "Technical Note", however, the number of figures seems excessive for this type of publication. I recommend the authors to consider reducing them. On my opinion, it is possible to remove a number of panels in Figs. 3-8 and merge the remaining panels. However, the complete figures for major reactions with all 12 panels for each month could be presented in Supporting Information if authors assume them important to a potential reader. In addition, Figure 11 could be moved in Supporting Information and Figure 12 removed at all. This modification may help better focus this work.

Following this remark and the remark by the Referee 1, we have organized the Supplement. In the revised manuscript, the Figures 3-8 have been reduced to Figures 3-4. However, the complete figures for the major reactions with all 12 panels for each month are presented in the Supplement. Figure 12 has been deleted. As a result, the number of Figures in the paper was reduced from 12 to 8, and the total number of panels in the paper was reduced from 143 to 75.

2. OH equilibrium boundary retrieving based on the data of Panka et al. (2021), presented in the Discussion, does not give much. The authors themselves say that most of these data do not allow us to determine the local altitude position of the boundary of the OH equilibrium region because they are cut off at 80 km. In addition, the authors have previously raised the question of the correctness of the Panka et al. approach for O retrieving below 85 km (that is based on proportionality atomic oxygen concentration to the ratio between the volume emission rates of OH* measured at 2.05 and 1.6 μm), see the discussion to the Kulikov et al. (2023, https://doi.org/10.5194/acp-23-14593-2023). It is therefore possible that the data of Panka et al. (2021) are in principle unsuitable for determining the OH equilibrium boundary. Although it is worth noting that even the small number of points presented in Fig. 12 show a seasonal and latitudinal dependence similar to the model curves in Fig. 10. However, I would suggest that this section be deleted and addressed in a separate paper along with the claimed reduction of O, H, OH, and $HO_2$ from SABER/TIMED data or some other data.

Following this remark and the remark by the Referee 1, we have excluded the OH equilibrium boundary retrieval with the Panka et al. data from the revised manuscript. Instead of deleted analysis, we give a detailed explanation in the Discussion, how the results of our paper can help to modify the Panka et al. method to extend its capabilities.

3. Lines 152-156. It seems to me that for the sake of clarity, it is necessary to specify where this statement comes from, and if we need accuracy not 10% but, for example, 1%, what would the criterion look like?

In the revised manuscript, this sentence was extended (see lines 166-169):

«Kulikov et al. (2023a) also showed, when $\tau_n \ll \tau_{n^{eq}}$, $n \cong n^{eq}(1 - sign(\frac{dn^{eq}}{dt}) \cdot \frac{\tau_n}{\tau_{n^{eq}}})$ in the first order approximation. Thus, the criterion

$$\tau_n/\tau_{n^{eq}} \leq 0.1 \tag{4}$$

is sufficient, in order to the possible relative difference between $n$ and $n^{eq}$ to be no more than 0.1.»

4. Lines 227-229. I think that in Supporting Information it would be appropriate to show the interesting features noted for the reactions H+O$_3$→OH+O$_2$ and HO$_2$+O→OH+O$_2$ at the 100-130 km altitudes.

Done. Please see Figure S12 and S13 in the Supplement.

5. Lines 424-427. The authors should add a couple of sentences clarifying their claim: "The simultaneous application of OH and HO$_2$ equilibrium conditions to the SABER data (O$_3$, volume emission rates at 2.0 and 1.6 μm) together with the criteria (16) and (24) to control this equilibrium validity is going to retrieve all unknown HOx - Ox components (O, H, OH, and HO$_2$), extending the altitude range of retrieval below 80 km and without external information."

In the revised manuscript, we give the detailed explanation to clarify this sense (see Discussion, lines 433-454):

«The results of our paper allow modifying the Panka et al. method to extend its capabilities. The simplest development of this method seems to be the following. First of all, note that the HO$_2$ equilibrium condition (9) depends on H and O only and can be used within the self-consistent retrieval procedure, considering the following system of equations:

$$OH(v = 1 - 9) = \frac{k_{12} \cdot H \cdot O_3 \cdot M \cdot f(v) + \sum_{v' > v}(a_1(v',v) + a_2(v',v) \cdot O_2 + a_3(v',v) \cdot N_2 + (a_4(v',v) + a_5(v',v)) \cdot O) \cdot OH(v')}{a_6(v)) \cdot O + \sum_{v > v'}(a_1(v,v') + a_2(v,v') \cdot O_2 + a_3(v,v') \cdot N_2 + (a_4(v,v') + a_5(v,v')) \cdot O)},$$

$$OH(0) = \frac{\sum_{v' > 0}(a_1(v',0) + a_2(v',0) \cdot O_2 + a_3(v',0) \cdot N_2 + (a_4(v',0) + a_5(v',0)) \cdot O) \cdot OH(v') + k_{18} \cdot O \cdot HO_2 + 2 \cdot k_{14} \cdot H \cdot HO_2}{k_{17} \cdot O},$$

$$HO_2 = \frac{k_{20} \cdot H \cdot M \cdot O_2}{k_{18} \cdot O + (k_{14} + k_{15} + k_{16}) \cdot H},$$

$$VER_{2\mu m} = a_1(9,7) \cdot OH(9) + a_1(8,9) \cdot OH(8),$$

$$VER_{1.6\mu m} = a_1(5,3) \cdot OH(5) + a_1(4,2) \cdot OH(4),$$

where $a_{1-6}$ are the constant rates of the processes $OH(v) \rightarrow OH(v'<v) + h\nu$, $OH(v) + O_2 \rightarrow OH(v<v') + O_2$, $OH(v) + N_2 \rightarrow OH(v<v') + N_2$, $OH(v) + O(^3P) \rightarrow OH(v' \leq v\text{-}5) + O(^1D)$, $OH(v) + O(^3P) \rightarrow OH(v'<v) + O(^3P)$, and $OH(v) + O(^3P) \rightarrow H + O_2$ respectively. Take into consideration, that this system includes 13 equations with 13 unknown variables. Therefore, the solution to the system for a single set of the SABER measurements (simultaneously measured profiles of $O_3$, T, pressure, $VER_{2\mu m}$, and $VER_{1.6\mu m}$) gives one simultaneously retrieved profiles of O, H, OH($v$=0-9), and $HO_2$. By applying the criteria (17) and (25) to obtained O and H profiles, we verify the fulfillment of OH and $HO_2$ equilibrium conditions and determine the height, below which the resulting profiles should be cut. More advanced retrieval procedure would be statistical, based on Bayesian theorem, taking into account the uncertainties in measurement data and rate constants. Similarly, for example, to Kulikov et al. (2018a), it should include a derivation of posterior conditional probability density function of retrieved characteristics and numerical analysis of this function. Detailed development of this retrieval method is outside of this paper and should be carried out in a separate work.»

6. I recommend to check the text of the work with the help of a professional editor, as I am not sure about the correctness of wording of some sentences.

The revised manuscript has been verified and corrected by a professional translator (American style).

Other changes are related to the recommendations and demands of other referees.

Thank you for taking your time to review our manuscript.

With respect,

Michael Kulikov, Michael Belikovich, Alexey Chubarov, Svetlana Dementyeva, and Alexander Feigin

---

## Author Response (AR2)

Dear Editor,

We would like to say many thanks the Referee for second review of our manuscript and providing valuable recommendations. We took into account all the remarks of Referee and, to the best of our ability, implemented the corresponding changes in the manuscript.

In the following, we address the comments point by point and show how the manuscript has been changed according to the comments. Below we use a certain color notation: comments by Referee are in red, our responses are in black, and the changes in the manuscript are in blue (placed inside the quotation marks).

**Response to the comments on the paper by Referee 2**

1. The paragraph at lines 390-401 is somewhat hard to follow. However, among other things, it seems to be saying that the summer/winter differences in O in MLT midlatitudes is due to photochemistry. There is ample evidence that this difference is driven primarily by the seasonal flip in the global-scale circulation. For example, see Wang et al. (2023a,b) and references therein. Wang, J. C., Yue, J., Wang, W., Qian, L., Wu, Q., & Wang, N. (2022). The lower thermospheric winter-to-summer meridional circulation: 1. Driving mechanism. Journal of Geophysical Research: Space Physics, 127, e2022JA030948. https://doi.org/10.1029/2022JA030948 Wang, J. C., Yue, J., Wang, W., Qian, L., Jones, M., Jr., & Wang, N. (2023). The lower thermospheric winter-to-summer meridional circulation: 2. Impact on atomic oxygen. Journal of Geophysical Research: Space Physics, 128, e2023JA031684. https://doi.org/10.1029/2023JA031684

In the revised manuscript, we have improved this paragraph to make it clearer (see lines 408-427 in Discussion):

« As noted, Figs. 5-6 represent an interesting peculiarity. At the middle latitudes summer $z_{HO_2{}_{sh}^{eq}}$ and $z_{OH_{sh}^{eq}}$ are remarkably higher than winter ones. For example, in February $z_{HO_2{}_{sh}^{eq}}$ at 60ºN is ~ 84 km, whereas the one at 60ºS is ~ 74 km. Recently, Kulikov et al. (2023b) found such a feature in the evolution of nighttime ozone chemical equilibrium boundary (Fig. 5 there), derived from SABER/TIMED data. The study showed that the boundary closely follows the transition zone that separates strong and weak diurnal oscillations of O and H (see Figs. 1-3 and 13 in Kulikov et al. (2023b)). Above the zone the behavior of components is dynamically driven and seasonality is the result of change in global-scale circulation, vertical advection being the primary factor according to Wang et al. (2023). In the transition zone and below O and H concentrations change by orders of magnitude during the night driven by photochemical processes. Kulikov et al. (2023b) studied the photochemistry at these altitudes and its seasonal dependence. It was shown analytically that

nighttime O decreases with the characteristic time scale $\tau_O = O/|dO/dt|$ proportional to the $O/H$ value at the beginning of the night (see Eq. (13) there). At the same time, according to the distributions derived from SABER measurements $O/H$ during summer daytime (and thus also at the beginning of the night) at the middle latitudes is remarkably less than the one during winter daytime (see Fig. 14 there). Consequently, summer values of nighttime $\tau_O$ below ~ 84 km are significantly shorter than winter ones, so summer O during the night decreases much faster than in winter. In our case lifetimes of $HO_2$ and OH are proportional mainly to $\frac{1}{O}$ (see Eqs. (11) and (19)), so, following the approach described in Section 2, the summer rise of $z_{HO_2{}_{sh}^{eq}}$ and $z_{OH_{sh}^{eq}}$ at the middle latitudes can be explained by the season difference in O diurnal photochemical evolution at these altitudes.»

Note we consider here the O diurnal evolution at ~74-84 km, where there are deep photochemical oscillations of O caused by the diurnal variations of solar radiation, when the difference between daytime and nighttime O values can reach several orders of magnitude. At this time, Wang et al. (2022, 2023) consider mainly the overlying region, where dynamical processes dominate. In particular, Wang et al. (2023) pointed «that the vertical advection is the dominant mechanism in redistributing O at altitudes between 84 and 103 km» (see Abstract there) or «Local vertical advection associated with the lower-thermospheric winter-to-summer meridional circulation is found to be the primary driver for redistributing O between 84 and 114 km» (see Conclusions there). The fact is duly noted in the new version of the paragraph.

2. Readers might find the ex post facto revision of critOH (lines 402-415) confusing. Why not introduce the correction at the point of the original derivation around line 340?

In the revised manuscript, this paragraph was moved to Section 6 immediately below the original derivation of $Crit_{OH}$ (see lines 351-363).

3. As far as I could see, the revised text has no mention of the presence of a supplemental document and does not refer to any specific figures from the supplement. Are these figures helpful in interpreting the results? Is this document even necessary?

The Supplement was organized due to requirements by other Referees (# 1 and 3). In the revised manuscript, we connected the figures from Supplement with the text of manuscript,

lines 234-235: «The complete figures for $HO_2$ sources and sinks for every month (all 12 panels) are given in Supplement (Figs. S3-S11)»

lines 257-259: «The complete figures for OH sources and sinks for every month (all 12 panels) are given in Supplement (Figs. S12-S24).»

lines 372-373: «(see Figs. S1-S2 and S25-S26 in Supplement)»

line 390: «(see Fig. S27 in Supplement)»

line 407: «(see Fig. S28 in Supplement)»

Thank you for taking your time to review our manuscript.

With respect,

Michael Kulikov, Michael Belikovich, Alexey Chubarov, Svetlana Dementyeva, and Alexander Feigin